# Newton-LESS: Sparsification without Trade-offs for the Sketched Newton Update

**Michał Dereziński**
Department of Statistics
University of California, Berkeley
mderezin@berkeley.edu

**Jonathan Lacotte**
Department of Electrical Engineering
Stanford University
lacotte@stanford.edu

**Mert Pilanci**
Department of Electrical Engineering
Stanford University
pilanci@stanford.edu

**Michael W. Mahoney**
ICSI and Department of Statistics
University of California, Berkeley
mmahoney@stat.berkeley.edu

## Abstract

In second-order optimization, a potential bottleneck can be computing the Hessian matrix of the optimized function at every iteration. Randomized sketching has emerged as a powerful technique for constructing estimates of the Hessian which can be used to perform approximate Newton steps. This involves multiplication by a random sketching matrix, which introduces a trade-off between the computational cost of sketching and the convergence rate of the optimization algorithm. A theoretically desirable but practically much too expensive choice is to use a dense Gaussian sketching matrix, which produces unbiased estimates of the exact Newton step and which offers strong problem-independent convergence guarantees. We show that the Gaussian sketching matrix can be drastically sparsified, significantly reducing the computational cost of sketching, without substantially affecting its convergence properties. This approach, called Newton-LESS, is based on a recently introduced sketching technique: LEverage Score Sparsified (LESS) embeddings. We prove that Newton-LESS enjoys nearly the same problem-independent local convergence rate as Gaussian embeddings, not just up to constant factors but even down to lower order terms, for a large class of optimization tasks. In particular, this leads to a new state-of-the-art convergence result for an iterative least squares solver. Finally, we extend LESS embeddings to include uniformly sparsified random sign matrices which can be implemented efficiently and which perform well in numerical experiments.

## 1 Introduction

Consider the task of minimizing a twice-differentiable convex function $f : \mathbb{R}^d \to \mathbb{R}$:

$$\text{find} \quad \mathbf{x}^* = \operatorname*{argmin}_{\mathbf{x} \in \mathbb{R}^d} f(\mathbf{x}).$$

One of the most classical iterative algorithms for solving this task is the Newton's method, which takes steps of the form $\mathbf{x}_{t+1} = \mathbf{x}_t - \mu_t \nabla^2 f(\mathbf{x}_t)^{-1} \nabla f(\mathbf{x}_t)$, and which leverages second-order information in the $d \times d$ Hessian matrix $\nabla^2 f(\mathbf{x}_t)$ to achieve rapid convergence, especially locally as it approaches the optimum $\mathbf{x}^*$. However, in many settings, the cost of forming the exact Hessian is prohibitively expensive, particularly when the function $f$ is given as a sum of $n \gg d$ components, i.e., $f(\mathbf{x}) = \sum_{i=1}^{n} f_i(\mathbf{x})$. This commonly arises in machine learning when $f$ represents the training loss over a

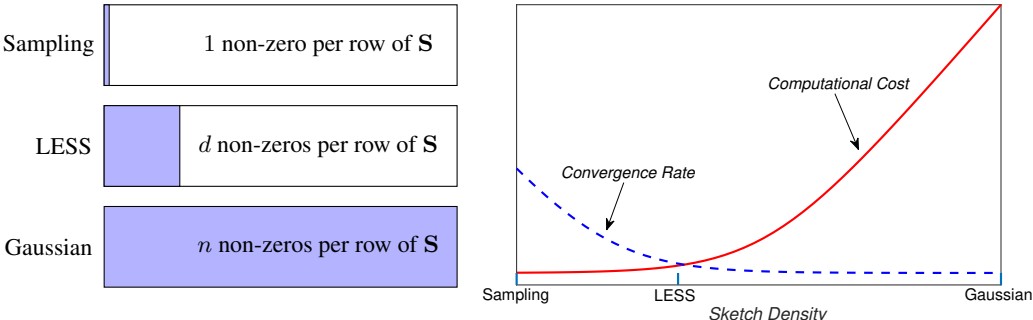

Figure 1: The effect of the density of $m \times n$ sketching matrix $\mathbf{S}$ applied to an $n \times d$ matrix $\mathbf{A}$ (with $d, m \ll n$) on the convergence rate of Newton Sketch and the computational cost of constructing the Hessian estimate. LESS embeddings "interpolate" between Sub-Sampled Newton methods and Gaussian Newton Sketches, achieving a "sweet spot" in the computation-per-iteration versus number-of-iterations tradeoff.

dataset of $n$ elements, as well as in solving semi-definite programs, portfolio optimization, and other tasks. In these contexts, we can represent the Hessian via a decomposition $\nabla^2 f(\mathbf{x}) = \mathbf{A}_f(\mathbf{x})^\top \mathbf{A}_f(\mathbf{x})$, where $\mathbf{A}_f(\mathbf{x})$ is a tall $n \times d$ matrix, which can be easily formed, and the main bottleneck is the matrix multiplication which takes $O(nd^2)$ arithmetic operations. To avoid this bottleneck, many randomized second-order methods have been proposed which use a Hessian estimate in place of the exact Hessian (e.g., [BCNN11, EM15, ABH17, RKM19]). This naturally leads to a trade-off between the per-iteration cost of the method and the number of iterations needed to reach convergence. We develop Newton-LESS, a randomized second-order method which eliminates the computational bottleneck while minimizing the convergence trade-offs.

An important family of approximate second-order methods is known as the Newton Sketch [PW17]:

$$\widetilde{\mathbf{x}}_{t+1} = \widetilde{\mathbf{x}}_t - \mu_t \big(\mathbf{A}_f(\widetilde{\mathbf{x}}_t)^\top \mathbf{S}_t^\top \mathbf{S}_t \mathbf{A}_f(\widetilde{\mathbf{x}}_t)\big)^{-1} \nabla f(\widetilde{\mathbf{x}}_t), \tag{1}$$

where $\mu_t$ is the step size, and $\mathbf{S}_t$ is a random $m \times n$ sketching matrix, with $m \ll n$, that is used to reduce $\mathbf{A}_f(\widetilde{\mathbf{x}}_t)$ to a small $m \times d$ sketch $\mathbf{S}_t \mathbf{A}_f(\widetilde{\mathbf{x}}_t)$. This brings the complexity of forming the Hessian down to $O(md^2)$ time plus the cost of forming the sketch.

Naturally, sketching methods vary in their computational cost and they can affect the convergence rate, so the right choice of $\mathbf{S}_t$ depends on the computation-convergence trade-off. On one end of this spectrum are the so-called Sub-Sampled Newton methods [RKM19, XRKM17, YXRKM18], where $\mathbf{S}_t$ simply selects a random sample of $m$ rows of $\mathbf{A}_f(\mathbf{x})$ (e.g., a sample of data points in a training set) to form the sketch. Here the sketching cost is negligible, since $\mathbf{S}_t$ is extremely sparse, but the convergence rate can be highly variable and problem-dependent. On the other end, we have what we will refer to as the Gaussian Newton Sketch, where $\mathbf{S}_t$ is a dense matrix with i.i.d. scaled Gaussian entries (a.k.a. a Gaussian embedding). While the $O(mnd)$ cost of performing this sketch limits its practical appeal, Gaussian Newton Sketch has a number of unique and desirable properties [LP19]: it enjoys strong problem-independent convergence rates; it produces unbiased estimates of the exact Newton update (useful in distributed settings); and it admits analytic expressions for the optimal step size.

A natural way to interpolate between these two extremes is to vary the sparsity $s$ of the sketching matrix $\mathbf{S}_t$, from $s = 1$ non-zero element per row (Sub-Sampling) to $s = n$ non-zero elements (Gaussian embedding), with the sketching complexity $O(mds)$.[1] Motivated by this, we ask:

> Can we sparsify the Gaussian embedding, making its sparsity closer to that of Sub-Sampling, without suffering *any* convergence trade-offs?

In this paper, we provide an affirmative answer to this question. We show that it is possible to drastically sparsify the Gaussian embedding so that two key statistics of the sketches, namely first

---

[1]For a more detailed discussion of other sketching techniques that may not fit this taxonomy, such as the Subsampled Randomized Hadamard Transform and the CountSketch, see Section 1.2.

and second inverse moments of the sketched Hessian, are nearly preserved in a very strong sense. Namely, the two inverse moments of the sparse sketches can be upper and lower bounded by the corresponding quantities for the dense Gaussian embeddings, where the upper/lower bounds are matching not just up to constant factors, but down to lower order terms (see Theorem 6). We use this to show that the Gaussian Newton Sketch can be sparsified to the point where the cost of sketching is proportional to the cost of other operations, while nearly preserving the convergence rate (again, down to lower order terms; see Theorem 1). This is illustrated conceptually in Figure 1, showing how the sparsity of the sketch affects the per-iteration convergence rate as well as the computational cost of the Newton Sketch. We observe that while the convergence rate improves as we increase the density, it eventually flattens out. On the other hand, the computational cost stays largely flat until some point when it starts increasing at a linear rate. As a result, there is a sparsity regime where we achieve the best of both worlds: the convergence rate and the computational cost are both nearly at their optimal values, thereby avoiding any trade-off.

To establish our results, we build upon a recently introduced sketching technique called LEverage Score Sparsified (LESS) embeddings [DLDM21]. LESS embeddings use leverage score techniques [DMIMW12a] to provide a carefully-constructed (random) sparsification pattern. This is used to produce a sub-Gaussian embedding with $d$ non-zeros per row of $\mathbf{S}_t$ (as opposed to $n$ for a dense matrix), so that the cost of forming the sketch $\mathbf{S}_t \mathbf{A}_f(\widetilde{\mathbf{x}}_t)$ matches the cost of constructing the Hessian estimate, i.e., $O(md^2)$ (see Section 2). [DLDM21] analyzed the first inverse moment of the sketch to show that LESS embeddings retain certain unbiasedness properties of Gaussian embeddings. In our setting, this captures the bias of the Newton Sketch, but it does not capture the variance, which is needed to control the convergence rate.

**Contributions.** In this paper, we analyze both the bias and the variance of Newton Sketch with LESS embeddings (Newton-LESS; see Definition 2 and Lemma 7), resulting in a comprehensive convergence analysis. The following are our key contributions:

1. Characterization of the second inverse moment of the sketched Hessian for a class of sketches including sub-Gaussian matrices and LESS embeddings;

2. Precise problem-independent local convergence rates for Newton-LESS, matching the Gaussian Newton Sketch down to lower order terms;

3. Extension of Newton-LESS to *regularized* minimization tasks, with improved dimension-independent guarantees for the sketch sparsity and convergence rate;

4. *Notable corollary:* Best known global convergence rate for an iterative least squares solver, which translates to state-of-the-art numerical performance.

## 1.1 Main results

As our main contribution, we show that, under standard assumptions on the function $f(\mathbf{x})$, Newton-LESS achieves the same problem-independent local convergence rate as the Gaussian Newton Sketch, despite drastically smaller per-iteration cost.

**Theorem 1.** *Assume that $f(\mathbf{x})$ is (a) self-concordant, or (b) has a Lipschitz continuous Hessian. Also, let $\mathbf{H} = \nabla^2 f(\mathbf{x}^*)$ be positive definite. There is a neighborhood $U$ containing $\mathbf{x}^*$ such that if $\widetilde{\mathbf{x}}_0 \in U$, then Newton-LESS with sketch size $m \geq Cd\log(dT/\delta)$ and step size $\mu_t = 1 - \frac{d}{m}$ satisfies:*

$$\left( \mathbb{E}_\delta \frac{\|\widetilde{\mathbf{x}}_T - \mathbf{x}^*\|_{\mathbf{H}}^2}{\|\widetilde{\mathbf{x}}_0 - \mathbf{x}^*\|_{\mathbf{H}}^2} \right)^{1/T} \approx_\epsilon \frac{d}{m} \quad for \quad \epsilon = O\left(\frac{1}{\sqrt{d}}\right),$$

*where $\mathbb{E}_\delta X$ is expectation conditioned on an event that holds with a $1 - \delta$ probability, $\|\mathbf{v}\|_{\mathbf{M}} = \sqrt{\mathbf{v}^\top \mathbf{M} \mathbf{v}}$, and $a \approx_\epsilon b$ means that $|a - b| \leq \epsilon b$.*

**Remark 2.** *The same guarantee holds for the Gaussian Newton Sketch, but it is not known for any fast sketching method other than Newton-LESS (see Section 1.2). The alternative assumptions of self-concordance and Lipschitz continuous Hessian are standard in the local convergence analysis of the classical Newton's method, and they only affect the size of the neighborhood $U$ (see Section 4). Global convergence of Newton-LESS follows from existing analysis of the Newton Sketch [PW17].*

The notion of expectation $\mathbb{E}_\delta$ allows us to accurately capture the average behavior of a randomized algorithm over a moderate (i.e., polynomial in $d$) number of trials even when the true expectation

is not well behaved. Here, this guards against the (very unlikely, but non-zero) possibility that the Hessian estimate produced by a sparse sketch will be ill-conditioned.

To illustrate this result in a special case (of obvious independent interest), we provide a simple corollary for the least squares regression task, i.e., $f(\mathbf{x}) = \frac{1}{2}\|\mathbf{Ax} - \mathbf{b}\|^2$. Importantly, here the convergence rate of $(\frac{d}{m})^T$ holds *globally*. Also, for this task we have $\frac{1}{2}\|\mathbf{x} - \mathbf{x}^*\|_{\mathbf{H}}^2 = f(\mathbf{x}) - f(\mathbf{x}^*)$, so the convergence can be stated in terms of the excess function value. To our knowledge, this is the best known convergence guarantee for a fast iterative least squares solver.

**Corollary 3.** *Let $f(\mathbf{x}) = \frac{1}{2}\|\mathbf{Ax} - \mathbf{b}\|^2$ for $\mathbf{A} \in \mathbb{R}^{n \times d}$ and $\mathbf{b} \in \mathbb{R}^n$. Then, given any $\widetilde{\mathbf{x}}_0 \in \mathbb{R}^d$, Newton-LESS with sketch size $m \geq Cd\log(dT/\delta)$ and step size $\mu_t = 1 - \frac{d}{m}$ satisfies:*

$$\left(\mathbb{E}_\delta \frac{f(\widetilde{\mathbf{x}}_T) - f(\mathbf{x}^*)}{f(\widetilde{\mathbf{x}}_0) - f(\mathbf{x}^*)}\right)^{1/T} \approx_\epsilon \frac{d}{m} \qquad for \quad \epsilon = O\left(\frac{1}{\sqrt{d}}\right).$$

Prior to this work, a convergence rate of $(\frac{d}{m})^T$ was known only for dense Gaussian embeddings, and only for the least squares task [LP19]. On the other hand, our results apply as generally as the standard local convergence analysis of the Newton's method, and they include a broad class of sketches. In Section 3, we provide general structural conditions on a randomized sketching matrix that are needed to enable our analysis. These conditions are satisfied by a wide range of sketching methods, including all sub-Gaussian embeddings (e.g., using random sign entries instead of Gaussians), the original LESS embeddings, and other choices of sparse random matrices (see Lemma 7). Moreover, we develop an improved local convergence analysis of the Newton Sketch, which allows us to recover the precise convergence rate and derive the optimal step size. In Appendix D, we also discuss a distributed variant of Newton-LESS, which takes advantage of the near-unbiasedness properties of LESS embeddings, extending the results of [DLDM21].

The performance of Newton-LESS can be further improved for regularized minimization tasks. Namely, suppose that function $f$ can be decomposed as follows: $f(\mathbf{x}) = f_0(\mathbf{x}) + g(\mathbf{x})$, where $g(\mathbf{x})$ has a Hessian that is easy to evaluate (e.g., $l_2$-regularization, $g(\mathbf{x}) = \frac{\lambda}{2}\|\mathbf{x}\|^2$). In this case, a modified variant of the Newton Sketch has been considered, where only the $f_0$ component is sketched:

$$\widetilde{\mathbf{x}}_{t+1} = \widetilde{\mathbf{x}}_t - \mu_t\left(\mathbf{A}_{f_0}(\widetilde{\mathbf{x}}_t)^\top \mathbf{S}_t^\top \mathbf{S}_t \mathbf{A}_{f_0}(\widetilde{\mathbf{x}}_t) + \nabla^2 g(\widetilde{\mathbf{x}}_t)\right)^{-1} \nabla f(\widetilde{\mathbf{x}}_t), \qquad (2)$$

where, again, we let $\mathbf{A}_{f_0}(\mathbf{x})$ be an $n \times d$ matrix that encodes the second-order information in $f_0$ at $\mathbf{x}$. For example, in the case of regularized least squares, $f(\mathbf{x}) = \frac{1}{2}\|\mathbf{Ax} - \mathbf{b}\|^2 + \frac{\lambda}{2}\|\mathbf{x}\|^2$, we have $\mathbf{A}_{f_0}(\mathbf{x}) = \mathbf{A}$ and $\nabla^2 g(\mathbf{x}) = \lambda\mathbf{I}$ for all $\mathbf{x}$. We show that the convergence rate of both Newton-LESS and the Gaussian Newton Sketch can be improved in the presence of regularization, by replacing the dimension $d$ with an *effective* dimension $d_{\text{eff}}$. This can be significantly smaller than $d$ when the Hessian of $f_0$ at the optimum exhibits rapid spectral decay or is approximately low-rank:

$$d_{\text{eff}} = \text{tr}\left(\nabla^2 f_0(\mathbf{x}^*)\nabla^2 f(\mathbf{x}^*)^{-1}\right) \leq d.$$

**Theorem 4.** *Assume that $f_0$ and $f$ are (a) self-concordant, or (b) have a Lipschitz continuous Hessian. Also, let $\nabla^2 f_0(\mathbf{x}^*)$ be positive definite and let $\nabla^2 g(\mathbf{x}^*)$ be positive semidefinite, with $\mathbf{H} = \nabla^2 f(\mathbf{x}^*)$. There is a neighborhood $U$ containing $\mathbf{x}^*$ such that if $\widetilde{\mathbf{x}}_0 \in U$, then Regularized Newton-LESS (2), with sketch size $m \geq Cd_{\text{eff}}\log(d_{\text{eff}}T/\delta)$ and step size $\mu_t = 1 - \frac{d_{\text{eff}}}{m}$, satisfies:*

$$\left(\mathbb{E}_\delta \frac{\|\widetilde{\mathbf{x}}_T - \mathbf{x}^*\|_{\mathbf{H}}^2}{\|\widetilde{\mathbf{x}}_0 - \mathbf{x}^*\|_{\mathbf{H}}^2}\right)^{1/T} \leq \frac{d_{\text{eff}}}{m} \cdot (1 + \epsilon) \qquad for \quad \epsilon = O\left(\frac{1}{\sqrt{d_{\text{eff}}}}\right).$$

**Remark 5.** *The same guarantee holds for the Gaussian Newton Sketch. Unlike in Theorem 1, here we can only obtain an upper-bound on the local convergence rate, because the exact rate may depend on the starting point $\widetilde{\mathbf{x}}_0$ (see Section 4). For regularized least squares, $f(\mathbf{x}) = \frac{1}{2}\|\mathbf{Ax} - \mathbf{b}\|^2 + \frac{\lambda}{2}\|\mathbf{x}\|^2$, the above convergence guarantee holds globally, i.e., $U = \mathbb{R}^d$. Note that $d_{\text{eff}}$ can be efficiently estimated using sketching-based trace estimators [ACW16, CEM$^+$15].*

Finally, our numerical results show that Newton-LESS can be implemented very efficiently on modern hardware platforms, improving on the optimization cost over not only dense Gaussian embeddings, but also state-of-the-art sketching methods such as the Subsampled Randomized Hadamard Transform, as well as other first-order and second-order methods. Moreover, we demonstrate that our theoretical predictions for the optimal sparsity level and convergence rate are extremely accurate in practice.

## 1.2 Related work

LEverage Score Sparsified (LESS) embeddings were proposed by [DLDM21] as a way of addressing the phenomenon of inversion bias, which arises in distributed second-order methods [WRKXM18, DM19, DBPM20, GGD$^+$21]. Their results only establish the *near-unbiasedness* of Newton-LESS iterates (i.e., that $\mathbb{E}[\widetilde{\mathbf{x}}_{t+1}] \approx \mathbf{x}_{t+1}$), but they did not provide any improved guarantees on the convergence rate. Also, their notion of LESS embeddings is much narrower than ours, and so it does not capture Regularized Newton-LESS or uniformly sparsified sketches (LESS-uniform).

Convergence analysis of the Newton Sketch [PW17, LP19, LLDP20] and other randomized second-order methods [BCNN11, BCNW12, EM15, RKM19] has been extensively studied in the machine learning community, often using techniques from Randomized Numerical Linear Algebra (RandNLA) [DM16, DM21]. Some of the popular RandNLA methods include the Subsampled Randomized Hadamard Transform (SRHT, [AC09]) and several variants of sparse sketches, such as the CountSketch [CW17, MM13] and OSNAP [NN13, Coh16]. Also, row sampling based on Leverage Scores [DMM06, AM15] and Determinantal Point Processes [Der19, DCV19, DM21] has been used for sketching. Note that CountSketch and OSNAP sparse sketches differ from LESS embeddings in several ways, and in particular, they use a fixed number of non-zeros per column of the sketching matrix (as opposed to per row), so unlike LESS, their rows are not independent. While all of the mentioned methods, when used in conjunction with the Newton Sketch, exhibit similar per-iteration complexity as LESS embeddings (see Section 2), their existing convergence analysis is fundamentally limited: The best known rate is $(C \log d \cdot \frac{d}{m})^T$, which is worse than our result of $(\frac{d}{m})^T$, by a factor of $C \log d$, where $C > 1$ is a non-negligible constant that arises in the measure concentration analysis.

In the specific context of least squares regression where $f(\mathbf{x}) = \frac{1}{2}\|\mathbf{A}\mathbf{x} - \mathbf{b}\|^2$, the Hessian $\mathbf{A}^\top \mathbf{A}$ remains constant, and an alternative strategy is to keep the random sketch $\mathbf{S}\mathbf{A}$ fixed at every iteration. Many efficient randomized iterative solvers are based on this precondition-and-solve approach [RT08, AMT10, MSM14]: form the sketch $\mathbf{S}\mathbf{A}$, compute an easy-to-invert square-root matrix $\tilde{\mathbf{H}}^{\frac{1}{2}}$ of $\mathbf{A}^\top \mathbf{S}^\top \mathbf{S}\mathbf{A}$ and apply an iterative least squares solver to the preconditioned objective $\min_{\mathbf{z}} \frac{1}{2}\|\mathbf{A}\tilde{\mathbf{H}}^{-\frac{1}{2}}\mathbf{z} - \mathbf{b}\|^2$, e.g., Chebyshev iterations or the preconditioned conjugate gradient. In contrast to the Newton Sketch, these methods do not naturally extend to more generic convex objectives for which the Hessian matrix changes at every iteration. Also, similarly as the Newton Sketch, their convergence guarantees are limited to $(C \log d \cdot \frac{d}{m})^T$ when used in conjunction with fast sketching methods such as SRHT, OSNAP, or leverage score sampling.

## 2 Preliminaries

**Notation.** We let $\|\mathbf{v}\|_{\mathbf{M}} = \sqrt{\mathbf{v}^\top \mathbf{M} \mathbf{v}}$. We define $a \approx_\epsilon b$ to mean $|a - b| \le \epsilon b$, whereas $a = b \pm \epsilon$ means that $|a - b| \le \epsilon$, and $C$ denotes a large absolute constant. We use $\mathbb{E}_{\mathcal{E}}$ to denote expectation conditioned on $\mathcal{E}$, and for a $\delta \in (0, 1)$, we use $\mathbb{E}_\delta$ as a short-hand for: "There is an event $\mathcal{E}$ with probability at least $1 - \delta$ s.t. $\mathbb{E}_{\mathcal{E}}$ ...". Let pd and psd mean positive definite and positive semidefinite. Random variable $X$ is sub-Gaussian if $\Pr\{|X| \ge t\} \le \exp(-ct^2)$ for all $t \ge 0$ and some $c = \Omega(1)$.

We next introduce some concepts related to LEverage Score Sparsified (LESS) embeddings. We start with the notion of statistical leverage scores [DMIMW12a], which are importance weights assigned to the rows of a matrix $\mathbf{A} \in \mathbb{R}^{n \times d}$. The definition below for leverage scores is somewhat more general than standard definitions, because it allows for a *regularized* leverage score, where the regularization depends on a $d \times d$ matrix $\mathbf{C}$. When $\mathbf{C}$ is a scaled identity matrix, this matches the definition of ridge leverage scores [AM15].

**Definition 1** (Leverage scores). *For given matrices $\mathbf{A} \in \mathbb{R}^{n \times d}$ and psd $\mathbf{C} \in \mathbb{R}^{d \times d}$, we define the ith leverage score $l_i(\mathbf{A}, \mathbf{C})$ as the squared norm of the ith row of $\mathbf{U} = \mathbf{A}\mathbf{H}^{-\frac{1}{2}}$, where $\mathbf{H} = \mathbf{A}^\top \mathbf{A} + \mathbf{C}$ is assumed to be invertible. The effective dimension of $\mathbf{A}$ (given $\mathbf{C}$) is defined as $d_{\text{eff}} = \sum_i l_i(\mathbf{A}, \mathbf{C}) = \text{tr}(\mathbf{U}^\top \mathbf{U})$, whereas the coherence of $\mathbf{A}$ (given $\mathbf{C}$) is $\tau = \frac{n}{d_{\text{eff}}} \max_i l_i(\mathbf{A}, \mathbf{C}) \in [1, \frac{n}{d_{\text{eff}}}]$.*

Next, we define a class of sparsified sub-Gaussian sketching matrices which will be used in our results. This captures LESS embeddings, as well as other sketching matrices that are supported by the analysis. To that end, we define what we call a *sparsifier*, which is an $n$-dimensional random vector $\boldsymbol{\xi}$ that specifies the sparsification pattern for one row of the $m \times n$ sketching matrix $\mathbf{S}$.

**Definition 2** (LESS embeddings). *Let $t_1, ..., t_s$ be sampled i.i.d. from a distribution $p = (p_1, ..., p_n)$. Then, the random vector $\boldsymbol{\xi}^\top = \left(\sqrt{\frac{b_1}{sp_1}}, ..., \sqrt{\frac{b_n}{sp_n}}\right)$, where $b_i = \sum_{j=1}^s 1_{[t_j=i]}$, is a $(p, s)$-sparsifier. A $(p, s)$-sparsified sub-Gaussian sketch is a random matrix $\mathbf{S}$ that consists of i.i.d. row vectors distributed as $c \cdot (\mathbf{x} \circ \boldsymbol{\xi})^\top$, where $\circ$ denotes the entry-wise product, $\mathbf{x}$ has i.i.d. mean zero, unit variance and sub-Gaussian entries, and $c$ is some constant. For given matrices $\mathbf{A} \in \mathbb{R}^{n \times d}$ and psd $\mathbf{C} \in \mathbb{R}^{d \times d}$, we focus on two variants of Leverage Score Sparsified embeddings:*

1. *LESS. We assume that $p_i \approx_{1/2} l_i(\mathbf{A}, \mathbf{C})/d_{\text{eff}}$, and let $s \approx_{1/2} d_{\text{eff}}$. For $\mathbf{C} = \mathbf{0}$ and $s = d$, we recover the LESS embeddings proposed by [DLDM21].*

2. *LESS-uniform. We simply let $p_i = 1/n$ (denoted as $p = \text{unif}$). This avoids the preprocessing needed for approximating the $l_i(\mathbf{A}, \mathbf{C})$, but we may need larger $s$ to recover the theory.*

**Computational cost.** To implement LESS, we must first approximate (i.e., there is no need to compute exactly [DMIMW12a]) the leverage scores of $\mathbf{A}$. This can be done in time $O(\text{nnz}(\mathbf{A}) \log n + d^3 \log d)$ by using standard RandNLA techniques [DMIMW12a, CW17], where $\text{nnz}(\mathbf{A})$ is the number of non-zero entries in $\mathbf{A}$ and it is bounded by $nd$. Since the prescribed sparsity for LESS satisfies $s = O(d)$, the sketching cost is at most $O(md^2)$. Thus, the total cost of constructing the sketched Hessian with LESS is $O(\text{nnz}(\mathbf{A}) \log n + md^2)$, which up to logarithmic factors matches other sparse sketching methods such as leverage score sampling (when implemented with approximate leverage scores [DMIMW12a]), CountSketch, and OSNAP. In comparison, using the SRHT leads to $O(nd \log m + md^2)$ complexity, since this method does not take advantage of data sparsity. Note that, in practice, the computational trade-offs between sketching methods are quite different, and significantly hardware-dependent (see Section 5). In particular, the cost of approximating the leverage scores in LESS embeddings can be entirely avoided by using LESS-uniform. Here, the total cost of sketching is $O(mds)$, but the sparsity of the sketch that is needed for the theory depends on $\mathbf{A}$ and $\mathbf{C}$. Yet, in Section 5, we show empirically that this approach works well even for $s = d$.

## 3 Equivalence between LESS and Gaussian Embeddings

In this section, we derive the basic quantities that determine the convergence properties of the Newton Sketch, namely, the first and second moments of the normalized sketched Hessian inverse. Our key technical contribution is a new analysis of the second moment for a wide class of sketching matrices that includes LESS embeddings and sub-Gaussian sketches.

Consider the Newton Sketch update as in (2), and let $\widetilde{\Delta}_t = \widetilde{\mathbf{x}}_t - \mathbf{x}^*$. Denoting $\mathbf{H}_t = \nabla^2 f(\widetilde{\mathbf{x}}_t)$, $\mathbf{g}_t = \nabla f(\widetilde{\mathbf{x}})$, and using $\mathbf{p}_t = -\mu_t \mathbf{H}_t^{-1} \mathbf{g}_t$ to denote the exact Newton direction with step size $\mu_t$, a simple calculation shows that:

$$\|\widetilde{\Delta}_{t+1}\|_{\mathbf{H}_t}^2 - \|\widetilde{\Delta}_t\|_{\mathbf{H}_t}^2 = 2\widetilde{\Delta}_t^\top \mathbf{H}_t^{\frac{1}{2}} \widetilde{\mathbf{Q}} \mathbf{H}_t^{\frac{1}{2}} \mathbf{p}_t + \mathbf{p}_t^\top \mathbf{H}_t^{\frac{1}{2}} \widetilde{\mathbf{Q}}^2 \mathbf{H}_t^{\frac{1}{2}} \mathbf{p}_t, \tag{3}$$

where $\widetilde{\mathbf{Q}} = \mathbf{H}_t^{\frac{1}{2}} (\mathbf{A}_{f_0}(\widetilde{\mathbf{x}}_t)^\top \mathbf{S}_t^\top \mathbf{S}_t \mathbf{A}_{f_0}(\widetilde{\mathbf{x}}_t) + \nabla^2 g(\widetilde{\mathbf{x}}_t))^{-1} \mathbf{H}_t^{\frac{1}{2}}$. From this, we have that the expected decrease in the optimization error is determined by the first two moments of the matrix $\widetilde{\mathbf{Q}}$, i.e., $\mathbb{E}[\widetilde{\mathbf{Q}}]$ and $\mathbb{E}[\widetilde{\mathbf{Q}}^2]$. In the unregularized case, i.e., $g(\mathbf{x}) = 0$, these moments can be derived exactly for the Gaussian embedding. For instance if we let $\mathbf{S}_t$ be an $m \times n$ matrix with i.i.d. standard normal entries scaled by $\frac{1}{\sqrt{m-d-1}}$, then we obtain that:

$$\mathbb{E}[\widetilde{\mathbf{Q}}] = \mathbf{I}, \qquad \mathbb{E}[\widetilde{\mathbf{Q}}^2] = \frac{(m-1)(m-d-1)}{(m-d)(m-d-3)} \cdot \mathbf{I} \approx_\epsilon \frac{m}{m-d} \cdot \mathbf{I},$$

for $\epsilon = O(1/d)$. This choice of scaling for the Gaussian Newton Sketch ensures that each iterate $\widetilde{\mathbf{x}}_{t+1}$ is an unbiased estimate of the corresponding exact Newton update with the same step size, i.e., that $\mathbb{E}[\widetilde{\mathbf{x}}_{t+1}] = \widetilde{\mathbf{x}}_t + \mathbf{p}_t$. For most other sketching techniques, neither of the two moments is analytically tractable because of the bias coming from matrix inversion. Moreover, if we allow for regularization, e.g., $g(\mathbf{x}) = \frac{\lambda}{2} \|\mathbf{x}\|^2$, then even the Gaussian embedding does not enjoy tractable formulas for the moments of $\widetilde{\mathbf{Q}}$. However, using ideas from asymptotic random matrix theory, [DLDM21] showed that in the unregularized case, the exact Gaussian formula for the first moment holds approximately for sub-Gaussian sketches and LESS embeddings: $\mathbb{E}_\delta[\widetilde{\mathbf{Q}}] \approx_\epsilon \mathbf{I}$. This implies near-unbiasedness of

the unregularized Newton-LESS iterates relative to the exact Newton step, but it is not sufficient to ensure any convergence guarantees.

In this work, we develop a general characterization of the first and second moments of $\widetilde{\mathbf{Q}}$ for a wide class of sketching matrices, both in the unregularized and in regularized settings. For the sake of generality, we will simplify the notation here, and analyze the first and second moment of $\mathbf{Q} = \mathbf{H}^{\frac{1}{2}}(\mathbf{A}^\top \mathbf{S}^\top \mathbf{S}\mathbf{A} + \mathbf{C})^{-1}\mathbf{H}^{\frac{1}{2}}$ for some matrices $\mathbf{A} \in \mathbb{R}^{n \times d}$ and $\mathbf{C} \in \mathbb{R}^{d \times d}$ such that $\mathbf{A}^\top \mathbf{A} + \mathbf{C} = \mathbf{H}$. In the context of Newton Sketch (2), these quantities correspond to $\mathbf{A} = \mathbf{A}_{f_0}(\widetilde{\mathbf{x}}_t)$ and $\mathbf{C} = \nabla^2 g(\widetilde{\mathbf{x}}_t)$. Also, as a shorthand, we will define the normalized version of matrix $\mathbf{A}$ as $\mathbf{U} = \mathbf{A}\mathbf{H}^{-\frac{1}{2}}$. The following are the two structural conditions that need to be satisfied by a sketching matrix to enable our analysis.

The first condition is standard in the sketching literature. Essentially, it implies that the sketching matrix $\mathbf{S}$ produces a useful approximation of the Hessian with high probability (although this guarantee is still far too coarse by itself to obtain our results).

**Condition 1** (Property of random matrix $\mathbf{S}$). *Given* $\mathbf{U} \in \mathbb{R}^{n \times d}$, *the* $m \times n$ *random matrix* $\mathbf{S}$ *satisfies* $\|\mathbf{U}^\top \mathbf{S}^\top \mathbf{S}\mathbf{U} - \mathbf{U}^\top \mathbf{U}\| \leq \eta$ *with probability* $1 - \delta$.

This property is known as the *subspace embedding property*. Subspace embeddings were first used by [DMM06], where they were used in a data-aware context to obtain relative-error approximations for $\ell_2$ regression and low-rank matrix approximation [DMM08]. Subsequently, data-oblivious subspace embeddings were used by [Sar06] and popularized by [Woo14]. Both data-aware and data-oblivious subspace embeddings can be used to derive bounds for the accuracy of various algorithms [DM16, DM18].

For our analysis, it is important to assume that $\mathbf{S}$ has i.i.d. row vectors $c\mathbf{s}_i^\top$, where $c$ is an appropriate scaling constant. The second condition is defined as a property of those row vectors, which makes them sufficiently similar to Gaussian vectors. This is a relaxation of the Restricted Bai-Silverstein condition, proposed by [DLDM21], which leads to significant improvements in the sparsity guarantee for LESS embeddings when the Newton Sketch is regularized.

**Condition 2** (Property of random vector $\mathbf{s}$). *Given* $\mathbf{U} \in \mathbb{R}^{n \times d}$, *the* $n$-*dimensional random vector* $\mathbf{s}$ *satisfies* $\mathrm{Var}[\mathbf{s}^\top \mathbf{U}\mathbf{B}\mathbf{U}^\top \mathbf{s}] \leq \alpha \cdot \mathrm{tr}(\mathbf{U}\mathbf{B}^2\mathbf{U}^\top)$ *for all p.s.d. matrices* $\mathbf{B}$ *and some* $\alpha = O(1)$.

Given these two conditions, we are ready to derive precise non-asymptotic analytic expressions for the first two moments of the regularized sketched inverse matrix, which is the main technical contribution of this work (proof in Appendix A).

**Theorem 6.** *Fix* $\mathbf{A}$ *and assume that* $\mathbf{C}$ *is psd. Define* $\mathbf{H} = \mathbf{A}^\top \mathbf{A} + \mathbf{C}$ *and* $\mathbf{U} = \mathbf{A}\mathbf{H}^{-\frac{1}{2}}$. *Let* $\mathbf{S}$ *consist of* $m$ *i.i.d. rows distributed as* $\frac{1}{\sqrt{m - d_{\mathrm{eff}}}}\mathbf{s}^\top$, *where* $\mathbb{E}[\mathbf{s}\mathbf{s}^\top] = \mathbf{I}_n$ *and* $d_{\mathrm{eff}} = \mathrm{tr}(\mathbf{U}^\top \mathbf{U})$. *Also, let* $\tilde{d}_{\mathrm{eff}} = \mathrm{tr}((\mathbf{U}^\top \mathbf{U})^2)$. *Suppose that the matrix consisting of the first* $m/3$ *rows of* $\mathbf{S}$ *scaled by* $\sqrt{3}$ *satisfies Condition 1 w.r.t.* $\mathbf{U}$, *for* $\eta \leq 1/2$ *and probability* $1 - \delta/3$, *where* $\delta \leq 1/m^3$. *Suppose also that* $\mathbf{s}$ *satisfies Condition 2 w.r.t.* $\mathbf{U}$. *If* $m \geq O(d_{\mathrm{eff}})$, *then, conditioned on event* $\mathcal{E}$ *that holds with probability* $1 - \delta$, *matrix* $\mathbf{Q} = \mathbf{H}^{\frac{1}{2}}(\mathbf{A}^\top \mathbf{S}^\top \mathbf{S}\mathbf{A} + \mathbf{C})^{-1}\mathbf{H}^{\frac{1}{2}}$ *satisfies* $\|\mathbf{Q} - \mathbf{I}\| \leq O(\eta)$ *and:*

$$\left\|\mathbb{E}_{\mathcal{E}}[\mathbf{Q}] - \mathbf{I}\right\| \leq O\left(\tfrac{\sqrt{d_{\mathrm{eff}}}}{m}\right), \qquad \left\|\mathbb{E}_{\mathcal{E}}[\mathbf{Q}^2] - \left(\mathbf{I} + \tfrac{d_{\mathrm{eff}}}{m - \tilde{d}_{\mathrm{eff}}}\mathbf{U}^\top \mathbf{U}\right)\right\| \leq O\left(\tfrac{\sqrt{d_{\mathrm{eff}}}}{m}\right).$$

Theorem 6 shows that, for a wide class of sketching matrices, we can approximately write $\mathbb{E}[\mathbf{Q}] \approx \mathbf{I}$ and $\mathbb{E}[\mathbf{Q}^2] \approx \mathbf{I} + \frac{d_{\mathrm{eff}}}{m - \tilde{d}_{\mathrm{eff}}}\mathbf{U}^\top \mathbf{U}$, with the error term scaling as $O\left(\frac{\sqrt{d_{\mathrm{eff}}}}{m}\right)$. In the case of the first moment, this is a relatively straightforward generalization of the unregularized formula for the Gaussian case. However, for the second moment this expression is considerably more complicated, including not one but two notions of effective dimension, $\tilde{d}_{\mathrm{eff}} \neq d_{\mathrm{eff}}$. To put this in context, in the unregularized case, i.e., $\mathbf{C} = \mathbf{0}$, we have $d_{\mathrm{eff}} = \tilde{d}_{\mathrm{eff}} = d$ and $\mathbf{U}^\top \mathbf{U} = \mathbf{I}$, so we get $\mathbf{I} + \frac{d_{\mathrm{eff}}}{m - \tilde{d}_{\mathrm{eff}}}\mathbf{U}^\top \mathbf{U} = \frac{m}{m - d}\mathbf{I}$, which matches the second moment for the Gaussian sketch (up to lower order terms).

In the case of the first moment, the proof of Theorem 6 follows along the same lines as in [DLDM21], using a decomposition of $\mathbb{E}[\mathbf{Q}]$ that is based on the Sherman-Morrison rank-one update of the inverse. This approach was originally inspired by the analysis of Stieltjes transforms that are used to establish the limiting spectral distribution in asymptotic random matrix theory (e.g., [BS10, CD11]), and applied to sketching by [DLLM20, DLDM21]. Our key contribution lies in deriving the bound for the second moment, which requires a substantially more elaborate decomposition of $\mathbb{E}[\mathbf{Q}^2]$.

In the following lemma, we establish that the assumptions of Theorem 6 are satisfied not only by Gaussian, but also sub-Gaussian and LESS embedding matrices (proof in Appendix C).

**Lemma 7.** *Fix $\mathbf{A}$ and assume that $\mathbf{C}$ is psd. Let $\mathbf{S}$ be a sketching matrix with $m$ i.i.d. rows distributed as $\frac{1}{\sqrt{m-d_{\mathrm{eff}}}}\mathbf{s}^{\top}$. Then, $\mathbf{S}$ satisfies Conditions 1 and 2 as long as one of the following holds:*

1. *Sub-Gaussian: $\mathbf{s}$ is an i.i.d. sub-Gaussian random vector and $m \geq C(d_{\mathrm{eff}} + \log(1/\delta))/\eta^2$;*

2. *LESS: $\mathbf{s}$ is a $(p, d_{\mathrm{eff}})$-sparsified i.i.d. sub-Gaussian random vector with $p_i \approx_{1/2} l_i(\mathbf{A}, \mathbf{C})/d_{\mathrm{eff}}$, and $m \geq Cd_{\mathrm{eff}} \log(d_{\mathrm{eff}}/\delta)/\eta^2$;*

3. *LESS-uniform: $\mathbf{s}$ is a $(\mathrm{unif}, \tau d_{\mathrm{eff}})$-sparsified i.i.d. sub-Gaussian random vector, where $\tau$ is the coherence of $\mathbf{A}$, i.e., $\frac{n}{d_{\mathrm{eff}}} \max_i l_i(\mathbf{A}, \mathbf{C})$, and $m \geq Cd_{\mathrm{eff}} \log(d_{\mathrm{eff}}/\delta)/\eta^2$.*

## 4 Convergence Analysis for Newton-LESS

In this section, we demonstrate how our main technical results can be used to provide improved convergence guarantees for Newton-LESS (and, more broadly, any sketching methods that satisfy the conditions of Theorem 6). Here, we will focus on the more general regularized setting (2), where we can only show an upper bound on the convergence rate (Theorem 4). The unregularized result (Theorem 1) with matching upper/lower bounds follows similarly.

To start, we introduce the standard assumptions on the function $f$, which are needed to ensure strong local convergence guarantees for the classical Newton's method [BV04].

**Assumption 8.** *Function $f : \mathbb{R}^d \to \mathbb{R}$ has a Lipschitz continuous Hessian with constant $L$, i.e., $\|\nabla^2 f(\mathbf{x}) - \nabla^2 f(\mathbf{x}')\| \leq L \|\mathbf{x} - \mathbf{x}'\|$ for all $\mathbf{x}, \mathbf{x}' \in \mathbb{R}^d$.*

**Assumption 9.** *Function $f : \mathbb{R}^d \to \mathbb{R}$ is self-concordant, i.e., for all $\mathbf{x}, \mathbf{x}' \in \mathbb{R}^d$, the function $\phi(t) = f(\mathbf{x} + t\mathbf{x}')$ satisfies: $|\phi'''(t)| \leq 2(\phi''(t))^{3/2}$.*

Only one of those two assumptions needs to be satisfied for our analysis to go through, and the choice of the assumption only affects the size of the neighborhood around the optimum $\mathbf{x}^*$ for which our local convergence guarantee is satisfied. To clarify this, below we give an expanded version of Theorem 4 (proof in Appendix B).

**Theorem 10** (Expanded Theorem 4)**.** *Let $\mathbf{H}_0 = \nabla^2 f_0(\mathbf{x}^*)$ be pd and $\mathbf{C} = \nabla^2 g(\mathbf{x}^*)$ be psd. Define $d_{\mathrm{eff}} = \mathrm{tr}(\mathbf{H}_0 \mathbf{H}^{-1})$ and $\tilde{d}_{\mathrm{eff}} = \mathrm{tr}((\mathbf{H}_0 \mathbf{H}^{-1})^2)$ for $\mathbf{H} = \mathbf{H}_0 + \mathbf{C}$. Assume one of the following:*

1. *$f_0$ and $f$ satisfy Assumption 8, and $U = \{\mathbf{x} : \|\mathbf{x} - \mathbf{x}^*\|_{\mathbf{H}} < \frac{\sqrt{d_{\mathrm{eff}}}}{m}(\lambda_{\min})^{3/2}/L\}$, where $\lambda_{\min}$ is the smallest eigenvalue of $\mathbf{H}_0$;*

2. *$f_0$ and $f$ satisfy Assumption 9, and $U = \{\mathbf{x} : \|\mathbf{x} - \mathbf{x}^*\|_{\mathbf{H}} < \frac{\sqrt{d_{\mathrm{eff}}}}{m}\}$.*

*Then, Newton Sketch (2) starting from $\widetilde{\mathbf{x}}_0 \in U$, using any $\mathbf{S}_t$ from Lemma 7 (i.e., sub-Gaussian, LESS, or LESS-uniform) with $\delta$ replaced by $\delta/T$, and step size $\mu_t = 1 - \frac{d_{\mathrm{eff}}}{m+d_{\mathrm{eff}}-\tilde{d}_{\mathrm{eff}}}$, satisfies:*

$$\left(\mathbb{E}_\delta \frac{\|\widetilde{\mathbf{x}}_T - \mathbf{x}^*\|_{\mathbf{H}}^2}{\|\widetilde{\mathbf{x}}_0 - \mathbf{x}^*\|_{\mathbf{H}}^2}\right)^{1/T} \leq \frac{d_{\mathrm{eff}} \cdot (1 + \epsilon)}{m + d_{\mathrm{eff}} - \tilde{d}_{\mathrm{eff}}} \leq \frac{d_{\mathrm{eff}}}{m} \cdot (1 + \epsilon), \qquad for \quad \epsilon = O\left(\frac{1}{\sqrt{d_{\mathrm{eff}}}}\right).$$

Note that, compared to Theorem 4, here we present a slightly sharper bound which uses both types of effective dimension, $\tilde{d}_{\mathrm{eff}} \leq d_{\mathrm{eff}}$, that are present in Theorem 6. The statement from Theorem 4 is recovered by replacing $\tilde{d}_{\mathrm{eff}}$ with $d_{\mathrm{eff}}$ in the step size and in the bound. The key step in the proof of the result is the following lemma, which uses Theorem 6 to characterize the Newton Sketch iterate $\widetilde{\mathbf{x}}_{t+1}$ in terms of the corresponding Newton iterate $\mathbf{x}_{t+1}$. Note that this result holds globally for arbitrary $\widetilde{\mathbf{x}}_t$ and without the smoothness assumptions on $f$. Recall that we let $\widetilde{\Delta}_t = \widetilde{\mathbf{x}}_t - \mathbf{x}^*$ denote the error residual at step $t$.

**Lemma 11.** *Fix $\mathbf{H}_t = \nabla^2 f(\widetilde{\mathbf{x}}_t)$ and let $\widetilde{\mathbf{x}}_{t+1}$ be the Newton Sketch iterate with $\mathbf{S}_t$ as in Lemma 7. If the exact Newton step $\mathbf{x}_{t+1} = \widetilde{\mathbf{x}}_t - \mu_t \mathbf{H}_t^{-1}\mathbf{g}_t$ is a descent direction, i.e., $\|\Delta_{t+1}\|_{\mathbf{H}_t} \leq \|\widetilde{\Delta}_t\|_{\mathbf{H}_t}$ where $\Delta_{t+1} = \mathbf{x}_{t+1} - \mathbf{x}^*$, then*

$$\mathbb{E}_\delta \|\widetilde{\Delta}_{t+1}\|_{\mathbf{H}_t}^2 = \|\Delta_{t+1}\|_{\mathbf{H}_t}^2 + \frac{d_{\mathrm{eff}}(\widetilde{\mathbf{x}}_t)}{m - \tilde{d}_{\mathrm{eff}}(\widetilde{\mathbf{x}}_t)}\|\mathbf{x}_{t+1} - \widetilde{\mathbf{x}}_t\|_{\nabla^2 f_0(\widetilde{\mathbf{x}}_t)}^2 \pm O\left(\frac{\sqrt{d_{\mathrm{eff}}}}{m}\right)\|\widetilde{\Delta}_t\|_{\mathbf{H}_t}^2.$$

Importantly, the second term on the right-hand side uses the norm $\|\cdot\|_{\nabla^2 f_0(\widetilde{\mathbf{x}}_t)}$, which is different than the norm $\|\cdot\|_{\mathbf{H}_t}$ used for the remaining terms. As a result, in the regularized setting, it is possible that the second term will be much smaller than the last term (the approximation error). This prevents us from obtaining a matching lower-bound for the convergence rate of Regularized Newton-LESS. On the other hand, when $g(\mathbf{x}) = 0$, then $f_0 = f$ and we obtain matching upper/lower bounds.

The remainder of the proof of Theorem 10 is essentially a variant of the local convergence analysis of the Newton's method. Here, note that typically, we would set the step size to $\mu_t = 1$ and we would expect superlinear (specifically, quadratic) convergence rate. However, for Newton Sketch, convergence is a mixture of linear rate and superlinear rate, where the linear part is due to the approximation error in sketching the Hessian. Sufficiently close to the optimum, the linear rate will dominate, and so this is what we focus on in our local convergence analysis. The key novelty here is that, unlike prior work, we strive to describe the linear rate precisely, down to lower order terms. As a key step, we observe that the convergence of exact Newton with step size $\mu_t < 1$, letting $\Delta_{t+1} = \mathbf{x}_{t+1} - \mathbf{x}^*$, is given by:

$$\|\Delta_{t+1}\|_{\mathbf{H}_t}^2 = \underbrace{(1-\mu_t)^2\|\widetilde{\Delta}_t\|_{\mathbf{H}_t}^2}_{\text{linear rate}} + \underbrace{\mu_t(\Delta_{t+1} + (1-\mu_t)\widetilde{\Delta}_t)^\top(\mathbf{H}_t\widetilde{\Delta}_t - \mathbf{g}_t)}_{\text{superlinear rate}}, \tag{4}$$

where recall that $\widetilde{\Delta}_t = \widetilde{\mathbf{x}}_t - \mathbf{x}^*$ corresponds to the previous iterate, and $\mathbf{g}_t = \nabla f(\widetilde{\mathbf{x}}_t)$. Here, $(1-\mu_t)^2$ represents the linear convergence rate. The superlinear term vanishes near the optimum $\mathbf{x}^*$, because of the presence of $\mathbf{H}_t\widetilde{\Delta}_t - \mathbf{g}_t$, which (under the smoothness assumptions on $f$) vanishes at the same rate as $\|\widetilde{\Delta}_t\|_{\mathbf{H}_t}^2$. Interestingly, with this precise analysis, the superlinear term does not have a quadratic rate, but rather a $3/2$ rate. Entering (4) into the guarantee from Lemma 11, we obtain that the local rate of Newton Sketch can be expressed as $(1-\mu_t)^2 + \frac{d_{\text{eff}}}{m-d_{\text{eff}}}\mu_t^2$, and minimizing this expression over $\mu_t$ we obtain the desired quantities from Theorem 10. Finally, we note that while the norms and effective dimensions in the above exact calculations are stated with respect to the Hessian $\mathbf{H}_t$ at the current iterate $\widetilde{\mathbf{x}}_t$, these can all be approximated by the corresponding quantities computed using the Hessian at the optimum $\mathbf{x}^*$ (as in Theorem 10), relying on smoothness of $f$.

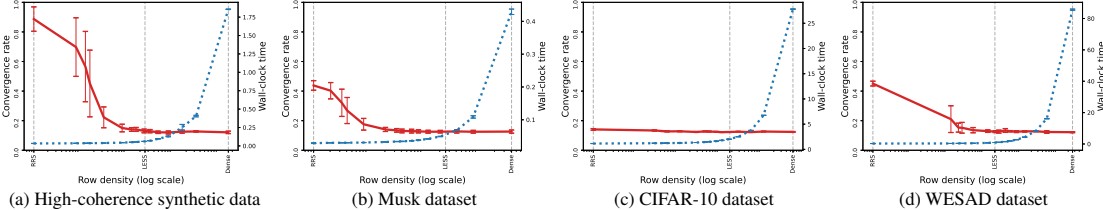

(a) High-coherence synthetic data     (b) Musk dataset     (c) CIFAR-10 dataset     (d) WESAD dataset

Figure 2: LESS-uniform embeddings: convergence rate of the Newton Sketch for least squares regression and wall-clock time of forming $\mathbf{SA}$ versus row density, with sketch size $m = 8d$. The results were averaged over 100 trials and error bars show twice the empirical standard deviation.

## 5  Numerical Experiments

We evaluated our theory on a range of different problems, and we have found that the more precise analysis that our theory provides describes well the convergence behavior for a range of optimization problems. In this section, we present numerical simulations illustrating this for regularized logistic regression and least squares regression, with different datasets ranging from medium to large scale: the CIFAR-10 dataset, the Musk dataset, and WESAD [SRD+18]. Data preprocessing and implementation details, as well as additional numerical results for least squares and regularized least squares, can be found in Appendix E.

We investigate first the effect of the row density of a LESS-uniform embedding on the Newton Sketch convergence rate and on the time for computing the sketch $\mathbf{SA}$ for least squares regression. In Figure 2, we report these two performance measures versus the row density. (This is the empirical analog for real data of Figure 1). Note that here we also consider a synthetic data matrix with high-coherence, which aims to be more challenging for a uniform sparsifier. Remarkably, our prescribed row density of $d$ non-zeros per row offers an excellent empirical trade-off: except for the CIFAR-10

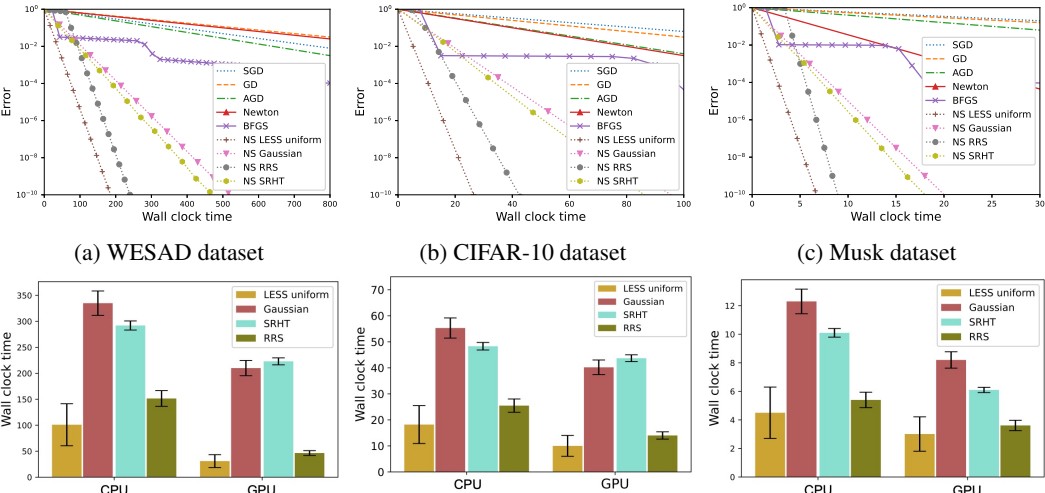

Figure 3: Top plots show the convergence of Newton Sketch (NS) and baselines for logistic regression. We use a sketch size $m = d/2$ for NS. In the bottom plots, we report the CPU and GPU wall-clock times to reach a $10^{-6}$ accurate solution for NS with different sketching methods.

dataset, for which random row sampling performs equally well to Gaussian embeddings, we observe that one can drastically decrease the row density without significantly impairing the convergence rate.

Next, in Figure 3, we investigate a minimization task for the regularized logistic regression loss. Namely, given a data matrix $\mathbf{A} \in \mathbb{R}^{n \times d}$ with rows $\mathbf{a}_i$, a target vector $\mathbf{b} \in \{\pm 1\}^n$ and a regularization parameter $\lambda > 0$, the goal is to solve:

$$\min_{\mathbf{x} \in \mathbb{R}^d} \frac{1}{n} \sum_{i=1}^{n} \log(1 + \exp(-b_i \mathbf{a}_i^\top \mathbf{x})) + \frac{\lambda}{2} \|\mathbf{x}\|_2^2.$$

For each dataset, we choose the value of $\lambda$ among $\{10^{-j} \mid j = 0, \dots, 8\}$ that minimizes the error on a hold out validation set. For CIFAR-10 and Musk, we pick $\lambda = 10^{-4}$. For WESAD, we pick $\lambda = 10^{-5}$. We plot the error versus wall-clock time for the Newton Sketch with LESS-uniform, Gaussian, Subsampled Randomized Hadamard Transform (SRHT) and Random Row Sampling (RRS) matrices, and we compare it with two standard second-order optimization baselines: Newton's method and BFGS. We also included three first-order baselines: Gradient Descent (GD), Accelerated GD (AGD), and Stochastic GD (SGD), observing much worse performance. Based on the top plots in Figure 3, we conclude that Newton-LESS (i.e., NS LESS uniform) offers significant time speed-ups over all baselines.

Finally, we compare wall-clock time on different hardwares (CPU versus GPU) for the Newton Sketch to reach a $10^{-6}$-accurate solution (see Appendix E for hardware details). From Figure 3 (bottom plots), we conclude the following: first, when switching from CPU to GPU for WESAD and CIFAR-10, Gaussian embeddings become more efficient than SRHT, despite a worse time complexity, by taking better advantage of the massively parallel architecture; second, Random Row Sampling performs better than either of them, despite having much weaker theoretical guarantees; and third, LESS-uniform is more efficient than all three other methods, on both CPU and GPU hardware platforms, observing a significant speed-up when switching to the parallel GPU architecture.

## 6 Conclusions

We showed that, when constructing randomized Hessian estimates for second-order optimization, we can get the best of both worlds: the efficiency of Sub-Sampling methods and the precision of Gaussian embeddings, by using sparse sketching matrices known as LEverage Score Sparsified (LESS) embeddings. Our algorithm, called Newton-LESS, enjoys both strong theoretical convergence guarantees and excellent empirical performance on a number of hardware platforms. An important future direction is to explain the surprising effectiveness of the simpler LESS-uniform method, particularly on high-coherence matrices, which goes beyond the predictions of our current theory.

**Acknowledgements**

We would like to acknowledge DARPA, IARPA, NSF, and ONR via its BRC on RandNLA for providing partial support of this work. MP and JL were partially supported by the National Science Foundation under grants IIS-1838179, ECCS-2037304 and the Army Research Office. Our conclusions do not necessarily reflect the position or the policy of our sponsors, and no official endorsement should be inferred.

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
