# A Characterization of Inverse Moments (Proof of Theorem 6)

In this section, we prove Theorem 6. The proof consists of two parts corresponding to the first and second moment of $\mathbf{Q}$. The analysis of the first moment bound is nearly the same as in [DLDM21], so we only outline it here, highlighting the differences coming from the regularization matrix $\mathbf{C}$. The analysis of the second moment is our main contribution in this proof, and we discuss it in detail. First, however, we define the high probability event $\mathcal{E}$ which is common to both parts.

To simplify the proof, we will let $m$ be divisible by 3. Note that we have $\mathbf{Q} = (\mathbf{U}^\top \mathbf{S}^\top \mathbf{S} \mathbf{U} + \mathbf{D})^{-1}$ for $\mathbf{D} = \mathbf{H}^{-\frac{1}{2}} \mathbf{C} \mathbf{H}^{-\frac{1}{2}}$. Moreover, let $\mathbf{S}_{-i}$ denote $\mathbf{S}$ without the $i$th row, and let $\mathbf{Q}_{-i} = (\mathbf{U}^\top \mathbf{S}_{-i}^\top \mathbf{S}_{-i} \mathbf{U} + \mathbf{D})^{-1}$. Also, define $\widetilde{\mathbf{S}}_1, \widetilde{\mathbf{S}}_2, \widetilde{\mathbf{S}}_3$ as the matrices consisting of the first, second and third group of $m/3$ rows in $\mathbf{S}$, all scaled by $\sqrt{3}$, so that $\mathbf{S}^\top \mathbf{S} = \frac{1}{3} \sum_{j=1}^3 \widetilde{\mathbf{S}}_j^\top \widetilde{\mathbf{S}}_j$. Next, using $\mathbf{\Sigma} = \mathbf{U}^\top \mathbf{U}$, similarly as in [DLDM21] we let $t = m/3$ and define three independent events:

$$\mathcal{E}_j : \quad \left\| \mathbf{\Sigma} - \mathbf{U}^\top \widetilde{\mathbf{S}}_j^\top \widetilde{\mathbf{S}}_j \mathbf{U} \right\| \leq \eta, \quad \text{for} \quad j = 1, 2, 3, \tag{5}$$

with $\mathcal{E} = \bigwedge_{j=1}^3 \mathcal{E}_j$ defined as the intersection of the events. Conditioned on $\mathcal{E}$, we have:

$$\|\mathbf{I} - (\mathbf{U}^\top \mathbf{S}^\top \mathbf{S} \mathbf{U} + \mathbf{D})\| = \left\| \frac{1}{3} \sum_{j=1}^3 \left( \mathbf{\Sigma} - \mathbf{U}^\top \widetilde{\mathbf{S}}_j^\top \widetilde{\mathbf{S}}_j \mathbf{U} \right) \right\| \leq \frac{1}{3} \sum_{j=1}^3 \|\mathbf{\Sigma} - \mathbf{U}^\top \widetilde{\mathbf{S}}_j^\top \widetilde{\mathbf{S}}_j \mathbf{U}\| \leq \eta,$$

which implies that $\|\mathbf{Q} - \mathbf{I}\| \leq \frac{\eta}{1-\eta} \leq 2\eta$. Furthermore, an important property of the definition of $\mathcal{E}$ is that for each $i \in \{1, ..., m\}$ there is a $j \in \{1, 2, 3\}$ such that $\mathcal{E}_j$ is independent of $\mathbf{x}_i$, and after conditioning only on $\mathcal{E}_j$ we get $\|\mathbf{Q}_{-i}\| \leq 6$. From Condition 1 and the union bound we conclude that $\Pr(\mathcal{E}) \geq 1 - \delta$.

The analysis of both the first and second moment uses the Sherman-Morrison formula, to separate one of the rows from the rest of the sketch. We state this formula in the following lemma.

**Lemma 12** (Sherman-Morrison). *For $\mathbf{A} \in \mathbb{R}^{n \times n}$ invertible and $\mathbf{u}, \mathbf{v} \in \mathbb{R}^n$, $\mathbf{A} + \mathbf{u} \mathbf{v}^\top$ is invertible if and only if $1 + \mathbf{v}^\top \mathbf{A}^{-1} \mathbf{u} \neq 0$ and*

$$(\mathbf{A} + \mathbf{u} \mathbf{v}^\top)^{-1} = \mathbf{A}^{-1} - \frac{\mathbf{A}^{-1} \mathbf{u} \mathbf{v}^\top \mathbf{A}^{-1}}{1 + \mathbf{v}^\top \mathbf{A}^{-1} \mathbf{u}}.$$

*From the above formula, it follows that:*

$$(\mathbf{A} + \mathbf{u} \mathbf{v}^\top)^{-1} \mathbf{u} = \frac{\mathbf{A}^{-1} \mathbf{u}}{1 + \mathbf{v}^\top \mathbf{A}^{-1} \mathbf{u}}.$$

## A.1 Proof of first moment bound

In this part of the proof we recall the decomposition of $\mathbb{E}_{\mathcal{E}}[\mathbf{Q}]$ used by [DLDM21]. Most of their analysis is unaffected by the presence of the regularization matrix $\mathbf{D}$, so we will focus on the steps that will also be needed for our analysis of the second moment. Let the $i$th row of $\mathbf{S}$ be $\frac{1}{\sqrt{m-d_{\text{eff}}}} \mathbf{s}_i^\top$, and define $\mathbf{x}_i = m \mathbf{U}^\top \mathbf{s}_i$, so that $\mathbf{U}^\top \mathbf{S}^\top \mathbf{S} \mathbf{U} = \frac{\gamma}{m} \sum_i \mathbf{x}_i \mathbf{x}_i^\top$, where $\gamma = \frac{m}{m-d_{\text{eff}}}$. Using $\gamma_i = 1 + \frac{\gamma}{m} \mathbf{x}_i^\top \mathbf{Q}_{-i} \mathbf{x}_i$, we have:

$$\begin{aligned}
\mathbb{E}_{\mathcal{E}}[\mathbf{Q}] - \mathbf{I} &= \mathbb{E}_{\mathcal{E}}[\mathbf{Q}(\mathbf{\Sigma} + \mathbf{D}) - \mathbf{Q}(\mathbf{U}^\top \mathbf{S}^\top \mathbf{S} \mathbf{U} + \mathbf{D})] \\
&= \mathbb{E}_{\mathcal{E}}[\mathbf{Q}\mathbf{\Sigma}] - \mathbb{E}_{\mathcal{E}}[\mathbf{Q}\mathbf{U}^\top \mathbf{S}^\top \mathbf{S} \mathbf{U}] \\
&\overset{(*)}{=} \mathbb{E}_{\mathcal{E}}[\mathbf{Q}\mathbf{\Sigma}] - \mathbb{E}_{\mathcal{E}}[\tfrac{\gamma}{\gamma_i} \mathbf{Q}_{-i} \mathbf{x}_i \mathbf{x}_i^\top] \\
&= \mathbb{E}_{\mathcal{E}}[\mathbf{Q} - \mathbf{Q}_{-i}]\mathbf{\Sigma} + \mathbb{E}_{\mathcal{E}}[\mathbf{Q}_{-i}(\mathbf{\Sigma} - \mathbf{x}_i \mathbf{x}_i^\top)] + \mathbb{E}_{\mathcal{E}}[(1 - \tfrac{\gamma}{\gamma_i})\mathbf{Q}_{-i} \mathbf{x}_i \mathbf{x}_i^\top],
\end{aligned}$$

where $(*)$ follows from the Sherman-Morrison formula. From this point, the analysis of [DLDM21] proceeds to bound the spectral norm of the first two terms by $O(1/m)$, and the spectral norm of the last term by $O(\sqrt{\text{tr}(\mathbf{U}^\top \mathbf{U})}/m)$. In their setup, $\mathbf{C} = \mathbf{0}$, which means that $\text{tr}(\mathbf{U}^\top \mathbf{U}) = d$, whereas in our more general statement, we let $d_{\text{eff}} = \text{tr}(\mathbf{U}^\top \mathbf{U})$. This does not affect the proofs. For the sake of our analysis of the second moment, we separate out the following guarantees obtained by [DLDM21], given here in a slightly more general form than originally.

**Lemma 13** ([DLDM21]). *The following bounds hold for $k \in \{1, 2\}$:*

$$\|\mathbb{E}_{\mathcal{E}}[\mathbf{Q}^k - \mathbf{Q}^k_{-i}]\mathbf{\Sigma}\| = O(1/m),$$

$$\|\mathbb{E}_{\mathcal{E}}[\mathbf{Q}^k_{-i}(\mathbf{x}_i\mathbf{x}_i^\top - \mathbf{\Sigma})]\| = O(1/m),$$

$$\|\mathbb{E}_{\mathcal{E}}[(\tfrac{\gamma}{\gamma_i} - 1)\mathbf{Q}^k_{-i}\mathbf{x}_i\mathbf{x}_i^\top]\| = O(\sqrt{d_{\text{eff}}}/m).$$

### A.2 Proof of second moment bound

We next present the analysis of the second moment, $\mathbb{E}_{\mathcal{E}}[\mathbf{Q}^2]$, which requires a considerably more elaborate decomposition. Using $\rho = \frac{d_{\text{eff}}}{m - d_{\text{eff}}}$ and $\mathbf{\Sigma} = \mathbf{U}^\top\mathbf{U}$, we have:

$$\mathbb{E}_{\mathcal{E}}[\mathbf{Q}^2] - (\mathbf{I} + \rho\mathbf{\Sigma}) = \underbrace{(\mathbb{E}_{\mathcal{E}}[\mathbf{Q}] - \mathbf{I})}_{\mathbf{T}_1} + (\mathbb{E}_{\mathcal{E}}[\mathbf{Q}(\mathbf{Q} - \mathbf{I})] - \rho\mathbf{\Sigma}).$$

Recalling that $\mathbf{\Sigma} + \mathbf{D} = \mathbf{I}$, we can rewrite the last term as:

$$
\begin{aligned}
\mathbb{E}_{\mathcal{E}}[\mathbf{Q}(\mathbf{Q} - \mathbf{I})] &= \mathbb{E}_{\mathcal{E}}[\mathbf{Q}(\mathbf{Q}(\mathbf{\Sigma} + \mathbf{D}) - \mathbf{Q}(\mathbf{U}^\top\mathbf{S}^\top\mathbf{S}\mathbf{U} + \mathbf{D}))] \\
&= \mathbb{E}_{\mathcal{E}}[\mathbf{Q}(\mathbf{Q}\mathbf{\Sigma} - \mathbf{Q}\mathbf{U}^\top\mathbf{S}^\top\mathbf{S}\mathbf{U})] \\
&\stackrel{(a)}{=} \mathbb{E}_{\mathcal{E}}[\mathbf{Q}(\mathbf{Q}\mathbf{\Sigma} - \tfrac{\gamma}{\gamma_i}\mathbf{Q}_{-i}\mathbf{x}_i\mathbf{x}_i^\top)] \\
&\stackrel{(b)}{=} \mathbb{E}_{\mathcal{E}}[\mathbf{Q}^2]\mathbf{\Sigma} - \mathbb{E}_{\mathcal{E}}[\tfrac{\gamma}{\gamma_i}\mathbf{Q}^2_{-i}\mathbf{x}_i\mathbf{x}_i^\top] + \mathbb{E}_{\mathcal{E}}\left[\tfrac{\mathbf{x}_i^\top\mathbf{Q}^2_{-i}\mathbf{x}_i}{m}\tfrac{\gamma^2}{\gamma_i^2}\mathbf{Q}_{-i}\mathbf{x}_i\mathbf{x}_i^\top\right] \\
&= \underbrace{\mathbb{E}_{\mathcal{E}}[\mathbf{Q}^2 - \mathbf{Q}^2_{-i}]\mathbf{\Sigma}}_{\mathbf{T}_2} + \underbrace{\mathbb{E}_{\mathcal{E}}[\mathbf{Q}^2_{-i}(\mathbf{\Sigma} - \mathbf{x}_i\mathbf{x}_i^\top)]}_{\mathbf{T}_3} + \underbrace{\mathbb{E}_{\mathcal{E}}[(1 - \tfrac{\gamma}{\gamma_i})\mathbf{Q}^2_{-i}\mathbf{x}_i\mathbf{x}_i^\top]}_{\mathbf{T}_4} + \mathbb{E}_{\mathcal{E}}\left[\tfrac{\mathbf{x}_i^\top\mathbf{Q}^2_{-i}\mathbf{x}_i}{m}\tfrac{\gamma^2}{\gamma_i^2}\mathbf{Q}_{-i}\mathbf{x}_i\mathbf{x}_i^\top\right],
\end{aligned}
$$

for a fixed $i$, where we denote $\gamma_i = 1 + \tfrac{\gamma}{m}\mathbf{x}_i^\top\mathbf{Q}_{-i}\mathbf{x}_i$. Note that we used the Sherman-Morrison formula twice, in steps $(a)$ and $(b)$. We can put everything together as follows:

$$
\begin{aligned}
\mathbb{E}_{\mathcal{E}}[\mathbf{Q}^2] - (\mathbf{I} + \rho\mathbf{\Sigma}) = \mathbf{T}_1 + \mathbf{T}_2 + \mathbf{T}_3 + \mathbf{T}_4 \\
+ \underbrace{\rho(\mathbb{E}_{\mathcal{E}}[\mathbf{Q}_{-i}] - \mathbf{I})\mathbf{\Sigma}}_{\mathbf{T}_5} + \underbrace{\rho\,\mathbb{E}_{\mathcal{E}}[\mathbf{Q}_{-i}(\mathbf{x}_i\mathbf{x}_i^\top - \mathbf{\Sigma})]}_{\mathbf{T}_6} + \underbrace{\mathbb{E}_{\mathcal{E}}\left[\left(\tfrac{\mathbf{x}_i^\top\mathbf{Q}^2_{-i}\mathbf{x}_i}{m}\tfrac{\gamma^2}{\gamma_i^2} - \rho\right)\mathbf{Q}_{-i}\mathbf{x}_i\mathbf{x}_i^\top\right]}_{\mathbf{T}_7}.
\end{aligned}
$$

From the bound on the first moment of $\mathbf{Q}$, we conclude that $\|\mathbf{T}_1\| = O(\sqrt{d_{\text{eff}}}/m)$ and that $\|\mathbf{T}_5\| = O(\sqrt{d_{\text{eff}}}/m)$. Without loss of generality, assume that events $\mathcal{E}_1$ and $\mathcal{E}_2$ are both independent of $\mathbf{x}_i$, and let $\mathcal{E}' = \mathcal{E}_1 \wedge \mathcal{E}_2$ as well as $\delta_3 = \Pr(\neg\mathcal{E}_3)$. Next, we will use the fact that for a p.s.d. random matrix $\mathbf{M}$ in the probability space of $\mathbf{S}$, we have $\mathbb{E}_{\mathcal{E}}[\mathbf{M}] \preceq \tfrac{1}{1-\delta}\mathbb{E}_{\mathcal{E}'}[\mathbf{M}] \preceq 2 \cdot \mathbb{E}_{\mathcal{E}'}[\mathbf{M}]$.

Using $k = 2$ in Lemma 13, we can bound $\|\mathbf{T}_2\|$, $\|\mathbf{T}_3\|$ and $\|\mathbf{T}_4\|$ by $O(\sqrt{d_{\text{eff}}}/m)$, and setting $k = 1$, we can do the same for $\|\mathbf{T}_6\|$.

Thus, it remains to bound $\|\mathbf{T}_7\|$. Let $\tilde{\gamma} = \frac{m}{m - \tilde{d}_{\text{eff}}}$. We first use the Cauchy-Schwartz inequality twice, obtaining that:

$$\|\mathbf{T}_7\| \le \frac{1}{m}\sqrt{\mathbb{E}_{\mathcal{E}}\left[\left(\tilde{\gamma}d_{\text{eff}} - \mathbf{x}_i^\top\mathbf{Q}^2_{-i}\mathbf{x}_i \cdot \tfrac{\gamma^2}{\gamma_i^2}\right)^2\right]} \cdot \sup_{\|\mathbf{u}\|=1}\sqrt[4]{\mathbb{E}_{\mathcal{E}}[(\mathbf{u}^\top\mathbf{Q}_{-i}\mathbf{x}_i)^4]} \cdot \sup_{\|\mathbf{u}\|=1}\sqrt[4]{\mathbb{E}_{\mathcal{E}}[(\mathbf{x}_i^\top\mathbf{u})^4]}.$$

(6)

The latter two terms can each be bounded easily by $O(\sqrt[4]{\alpha + 1})$ using Condition 2. For instance, considering the middle term, we have:

$$
\begin{aligned}
\mathbb{E}_{\mathcal{E}}[(\mathbf{u}^\top\mathbf{Q}_{-i}\mathbf{x}_i)^4] &\le 2\,\mathbb{E}_{\mathcal{E}'}\left[\mathbb{E}[(\mathbf{x}_i^\top\mathbf{Q}_{-i}\mathbf{u}\mathbf{u}^\top\mathbf{Q}_{-i}\mathbf{x}_i)^2 \mid \mathbf{Q}_{-i}]\right] \\
&= 2\,\mathbb{E}_{\mathcal{E}'}\left[\text{Var}[\mathbf{x}_i^\top\mathbf{Q}_{-i}\mathbf{u}\mathbf{u}^\top\mathbf{Q}_{-i}\mathbf{x}_i \mid \mathbf{Q}_{-i}] + (\mathbb{E}[\mathbf{x}_i^\top\mathbf{Q}_{-i}\mathbf{u}\mathbf{u}^\top\mathbf{Q}_{-i}\mathbf{x}_i \mid \mathbf{Q}_{-i}])^2\right] \\
&\le 2\,\mathbb{E}_{\mathcal{E}'}\left[\alpha\,\text{tr}(\mathbf{U}(\mathbf{Q}_{-i}\mathbf{u}\mathbf{u}^\top\mathbf{Q}_{-i})^2\mathbf{U}^\top) + 2\,(\text{tr}(\mathbf{U}\mathbf{Q}_{-i}\mathbf{u}\mathbf{u}^\top\mathbf{Q}_{-i}\mathbf{U}^\top))^2\right] \\
&\le 2\,\mathbb{E}_{\mathcal{E}'}\left[O(\alpha)\,\mathbf{u}_i^\top\mathbf{Q}_{-i}\mathbf{U}^\top\mathbf{U}\mathbf{Q}_{-i}\mathbf{u}_i + (\mathbf{u}_i^\top\mathbf{Q}_{-i}\mathbf{U}^\top\mathbf{U}\mathbf{Q}_{-i}\mathbf{u}_i)^2\right] \\
&\le 2\,\mathbb{E}_{\mathcal{E}'}\left[O(\alpha)\|\mathbf{Q}_{-i}\mathbf{\Sigma}\mathbf{Q}_{-i}\| + \|\mathbf{Q}_{-i}\mathbf{\Sigma}\mathbf{Q}_{-i}\|^2\right] = O(\alpha + 1),
\end{aligned}
$$

where we also used that matrices $\mathbf{Q}_{-i}$, $\boldsymbol{\Sigma}$ and $\mathbf{u}\mathbf{u}^\top$ have spectral norms bounded by $O(1)$. Similarly, we obtain that $\mathbb{E}_{\mathcal{E}}[(\mathbf{x}_i^\top \mathbf{u})^4] = O(\alpha + 1)$. For the first term in (6), we have:

$$\mathbb{E}_{\mathcal{E}}\left[\left(\tilde{\gamma}d_{\text{eff}} - \mathbf{x}_i^\top \mathbf{Q}_{-i}^2 \mathbf{x}_i \cdot \tfrac{\gamma^2}{\gamma_i^2}\right)^2\right] \leq 2 \cdot \mathbb{E}_{\mathcal{E}'}\left[\left(\tilde{\gamma}d_{\text{eff}} - \mathbf{x}_i^\top \mathbf{Q}_{-i}^2 \mathbf{x}_i \cdot \tfrac{\gamma^2}{\gamma_i^2}\right)^2\right]$$

$$\leq 4 \cdot \mathbb{E}_{\mathcal{E}'}\left[(\tilde{\gamma}d_{\text{eff}} - \mathbf{x}_i^\top \mathbf{Q}_{-i}^2 \mathbf{x}_i)^2\right] + 4 \cdot \mathbb{E}_{\mathcal{E}'}\left[\left(\mathbf{x}_i^\top \mathbf{Q}_{-i}^2 \mathbf{x}_i\right)^2\left(\tfrac{\gamma^2}{\gamma_i^2} - 1\right)^2\right].$$

We can further break down the first term as follows:

$$\mathbb{E}_{\mathcal{E}'}\left[(\tilde{\gamma}d_{\text{eff}} - \mathbf{x}_i^\top \mathbf{Q}_{-i}^2 \mathbf{x}_i)^2\right] = (\tilde{\gamma}d_{\text{eff}} - \mathbb{E}_{\mathcal{E}'}[\text{tr}(\mathbf{Q}_{-i}^2\boldsymbol{\Sigma})])^2 + \text{Var}_{\mathcal{E}'}[\text{tr}(\mathbf{Q}_{-i}^2\boldsymbol{\Sigma})] + \mathbb{E}_{\mathcal{E}'}\left[(\text{tr}(\mathbf{Q}_{-i}^2\boldsymbol{\Sigma}) - \mathbf{x}_i^\top \mathbf{Q}_{-i}^2\mathbf{x}_i)^2\right]$$
$$(7)$$

The latter term can be bounded immediately using Condition 2. The middle term is handled by a separate lemma, which is an immediate extension of Lemma 25 in [DLDM21].

**Lemma 14** ([DLDM21]). *Let* $\text{Var}_{\mathcal{E}'}[\cdot]$ *be the conditional variance with respect to event* $\mathcal{E}' = \mathcal{E}_1 \wedge \mathcal{E}_2$. *Then, for* $k \in \{1, 2\}$,

$$\text{Var}_{\mathcal{E}'}\left[\text{tr}(\mathbf{Q}_{-i}^k\boldsymbol{\Sigma})\right] = O(d_{\text{eff}}).$$

Next, note that $|\tfrac{\gamma^2}{\gamma_i^2} - 1| = |\gamma - \gamma_i| \cdot \tfrac{\gamma + \gamma_i}{\gamma_i^2} \leq |\gamma - \gamma_i| \cdot \tfrac{\gamma + 1}{\gamma_i}$, since $\gamma_i > 1$, so we get:

$$\mathbb{E}_{\mathcal{E}'}\left[\left(\mathbf{x}_i^\top \mathbf{Q}_{-i}^2 \mathbf{x}_i\right)^2\left(\tfrac{\gamma^2}{\gamma_i^2} - 1\right)^2\right] \leq 6^2(\gamma + 1)^2 \cdot \mathbb{E}_{\mathcal{E}'}\left[\left(\mathbf{x}_i^\top \mathbf{Q}_{-i}\mathbf{x}_i\right)^2 \tfrac{(\gamma - \gamma_i)^2}{\gamma_i^2}\right]$$

$$\leq 6^2(\gamma + 1)^2 \cdot \mathbb{E}_{\mathcal{E}'}\left[\tfrac{(\mathbf{x}_i^\top \mathbf{Q}_{-i}\mathbf{x}_i)^2}{(1 + \tfrac{\gamma}{m}\mathbf{x}_i^\top \mathbf{Q}_{-i}\mathbf{x}_i)^2}(\gamma - \gamma_i)^2\right]$$

$$\leq O(m^2) \cdot \mathbb{E}_{\mathcal{E}'}\left[(\gamma - \gamma_i)^2\right]$$

$$\leq O(m^2) \cdot O(\alpha d_{\text{eff}}/m^2) = O(\alpha d_{\text{eff}}).$$

Finally, we analyze the first term in (7) as follows:

$$\left|\tilde{\gamma}d_{\text{eff}} - \mathbb{E}_{\mathcal{E}'}[\text{tr}(\mathbf{Q}_{-i}^2\boldsymbol{\Sigma})]\right| = \left|\text{tr}((\mathbb{E}_{\mathcal{E}} - \mathbb{E}_{\mathcal{E}'})[\mathbf{Q}_{-i}^2\boldsymbol{\Sigma}]) - \text{tr}(\mathbf{T}_2) + \text{tr}(\tilde{\gamma}\boldsymbol{\Sigma} - \mathbb{E}_{\mathcal{E}}[\mathbf{Q}^2]\boldsymbol{\Sigma})\right|$$

$$= \left|\text{tr}((\mathbb{E}_{\mathcal{E}} - \mathbb{E}_{\mathcal{E}'})[\mathbf{Q}_{-i}^2\boldsymbol{\Sigma}]) - \text{tr}(\mathbf{T}_2) + \text{tr}((\mathbf{I} + \rho\boldsymbol{\Sigma} - \mathbb{E}_{\mathcal{E}}[\mathbf{Q}^2])\boldsymbol{\Sigma})\right|$$

$$\leq O(d_{\text{eff}}/m^3) + d_{\text{eff}} \cdot O(\alpha\sqrt{d_{\text{eff}}}/m) + |\text{tr}(\mathbf{T}_7\boldsymbol{\Sigma})|,$$

where to bound the first term we used the fact that $\Pr(\neg\mathcal{E}_3 \mid \mathcal{E}') \leq 1/m^3$ and for the last term, recall that $d_{\text{eff}} = \text{tr}(\boldsymbol{\Sigma})$ and $\tilde{d}_{\text{eff}} = \text{tr}(\boldsymbol{\Sigma}^2)$, which leads to the following identity:

$$\text{tr}(\tilde{\gamma}\boldsymbol{\Sigma} - (\mathbf{I} + \rho\boldsymbol{\Sigma})\boldsymbol{\Sigma}) = \text{tr}\left(\tfrac{\tilde{d}_{\text{eff}}}{m - \tilde{d}_{\text{eff}}}\boldsymbol{\Sigma} - \tfrac{d_{\text{eff}}}{m - \tilde{d}_{\text{eff}}}\boldsymbol{\Sigma}^2\right) = 0.$$

Further, note that from the analysis of $\mathbf{T}_7$ we have:

$$|\text{tr}(\mathbf{T}_7)| \leq \tfrac{d_{\text{eff}}}{m}\sqrt{4\left(\tilde{\gamma}d_{\text{eff}} - \mathbb{E}_{\mathcal{E}'}[\text{tr}(\mathbf{Q}_{-i}^2)]\right)^2 + O(\alpha d_{\text{eff}})} \cdot O(\sqrt{\alpha})$$

$$\leq O(\alpha d_{\text{eff}}/m) \cdot \left(|\tilde{\gamma}d_{\text{eff}} - \mathbb{E}_{\mathcal{E}'}[\text{tr}(\mathbf{Q}_{-i}^2)]| + \sqrt{d_{\text{eff}}}\right).$$

Putting this together with the previous inequality, we conclude that for sufficiently large $m$:

$$\left|\tilde{\gamma}d_{\text{eff}} - \mathbb{E}_{\mathcal{E}'}[\text{tr}(\mathbf{Q}_{-i}^2)]\right| \leq \frac{O(\alpha\sqrt{d_{\text{eff}}})}{1 - O(\alpha d_{\text{eff}}/m)} = O(\alpha\sqrt{d_{\text{eff}}}).$$

Plugging this back into the analysis of $\|\mathbf{T}_7\|$, we can bound it by $O(\alpha\sqrt{d}/m)$, which concludes the proof.

## B Local Convergence Rate of Newton-LESS

In this section, we present the convergence analysis of Newton Sketch for sketching matrices satisfying the structural conditions of Theorem 6. We start by proving Lemma 11, then we show how it can be used to establish the guarantee from Theorem 10. Finally, we discuss how the analysis needs to be adjusted to obtain the two-sided bound from Theorem 1.

## B.1 Proof of Lemma 11

Let $\widetilde{\Delta}_t = \widetilde{\mathbf{x}}_t - \mathbf{x}^*$, $\Delta_{t+1} = \mathbf{x}_{t+1} - \mathbf{x}^*$, and $\mathbf{p}_t = \mathbf{x}_{t+1} - \mathbf{x}_t$. Also, define $\rho = \frac{d_{\text{eff}}}{m - d_{\text{eff}}}$ as well as the matrices $\widetilde{\mathbf{Q}} = \mathbf{H}_t^{\frac{1}{2}} (\mathbf{A}_{f_0}(\widetilde{\mathbf{x}}_t)^\top \mathbf{S}_t^\top \mathbf{S}_t \mathbf{A}_{f_0}(\widetilde{\mathbf{x}}_t) + \nabla^2 g(\widetilde{\mathbf{x}}_t))^{-1} \mathbf{H}_t^{\frac{1}{2}}$ and $\mathbf{U}_t = \mathbf{A}_{f_0}(\widetilde{\mathbf{x}}) \mathbf{H}_t^{-\frac{1}{2}}$. We have:

$$
\begin{aligned}
\mathbb{E}_{\mathcal{E}} \|\widetilde{\Delta}_{t+1}\|_{\mathbf{H}_t}^2 - \|\Delta_{t+1}\|_{\mathbf{H}_t}^2 &= 2\Delta_{t+1}^\top \mathbf{H}_t \mathbb{E}_{\mathcal{E}} [\widetilde{\mathbf{x}}_{t+1} - \mathbf{x}_{t+1}] + \mathbb{E}_{\mathcal{E}} \|\widetilde{\mathbf{x}}_{t+1} - \mathbf{x}_{t+1}\|_{\mathbf{H}_t}^2 \\
&= 2\Delta_{t+1}^\top \mathbf{H}_t^{\frac{1}{2}} (\mathbf{I} - \mathbb{E}_{\mathcal{E}} \widetilde{\mathbf{Q}}) \mathbf{H}_t^{\frac{1}{2}} \mathbf{p}_t + \mathbf{p}_t^\top \mathbf{H}_t^{\frac{1}{2}} \mathbb{E}_{\mathcal{E}} (\mathbf{I} - \widetilde{\mathbf{Q}})^2 \mathbf{H}_t^{\frac{1}{2}} \mathbf{p}_t \\
&\leq \rho \cdot \mathbf{p}_t^\top \mathbf{H}_t^{\frac{1}{2}} \mathbf{U}_t^\top \mathbf{U}_t \mathbf{H}_t^{\frac{1}{2}} \mathbf{p}_t + 2\|\mathbf{I} - \mathbb{E}_{\mathcal{E}} \widetilde{\mathbf{Q}}\| \cdot \left(\|\Delta_{t+1}\|_{\mathbf{H}_t} \|\mathbf{p}_t\|_{\mathbf{H}_t} + \|\mathbf{p}_t\|_{\mathbf{H}_t}^2\right) \\
&\quad + \|\mathbf{I} + \rho \mathbf{U}_t^\top \mathbf{U}_t - \mathbb{E}_{\mathcal{E}} \widetilde{\mathbf{Q}}^2\| \cdot \|\mathbf{p}_t\|_{\mathbf{H}_t}^2 \\
&\leq \rho \|\mathbf{p}_t\|_{\nabla^2 f_0(\widetilde{\mathbf{x}}_t)}^2 + O\big(\tfrac{\sqrt{d_{\text{eff}}}}{m}\big) \|\widetilde{\Delta}_t\|_{\mathbf{H}_t}^2,
\end{aligned}
$$

where the last step follows by applying Theorem 6 and observing that $\|\mathbf{p}_t\|_{\mathbf{H}_t} \leq \|\widetilde{\Delta}_t\|_{\mathbf{H}_t} + \|\Delta_{t+1}\|_{\mathbf{H}_t} \leq 2\|\widetilde{\Delta}_t\|_{\mathbf{H}_t}$ and $\|\Delta_{t+1}\|_{\mathbf{H}_t} \|\mathbf{p}_t\|_{\mathbf{H}_t} \leq \frac{1}{2}(\|\Delta_{t+1}\|_{\mathbf{H}_t}^2 + \|\mathbf{p}_t\|_{\mathbf{H}_t}^2) \leq 3\|\widetilde{\Delta}_t\|_{\mathbf{H}_t}^2$. The matching lower-bound follows identically.

## B.2 Proof of Theorem 10

We start by analyzing the exact Newton step $\mathbf{x}_{t+1} = \widetilde{\mathbf{x}}_t - \mu_t \mathbf{H}_t^{-1} \mathbf{g}_t$ wih step size $\mu_t$, gradient $\mathbf{g}_t = \nabla f(\widetilde{\mathbf{x}}_t)$, and Hessian $\mathbf{H}_t = \nabla^2 f(\widetilde{\mathbf{x}}_t)$. Letting $\widetilde{\Delta}_t = \widetilde{\mathbf{x}}_t - \mathbf{x}^*$ and $\Delta_{t+1} = \mathbf{x}_{t+1} - \mathbf{x}^*$, we have:

$$
\begin{aligned}
\|\Delta_{t+1}\|_{\mathbf{H}_t}^2 &= (1 - \mu_t) \Delta_{t+1}^\top \mathbf{g}_t + \Delta_{t+1}^\top (\mathbf{H}_t \widetilde{\Delta}_t - \mathbf{g}_t) \\
&= (1 - \mu_t) \Delta_{t+1}^\top \mathbf{H}_t \widetilde{\Delta}_t - (1 - \mu_t) \Delta_{t+1}^\top (\mathbf{H}_t \widetilde{\Delta}_t - \mathbf{g}_t) + \Delta_{t+1}^\top (\mathbf{H}_t \widetilde{\Delta}_t - \mathbf{g}_t) \\
&= (1 - \mu_t) \big(\widetilde{\Delta}_t^\top \mathbf{H}_t \widetilde{\Delta}_t - \mu_t \mathbf{g}_t^\top \widetilde{\Delta}_t\big) + \mu_t \Delta_{t+1}^\top (\mathbf{H}_t \widetilde{\Delta}_t - \mathbf{g}_t) \\
&= (1 - \mu_t)^2 \|\widetilde{\Delta}_t\|_{\mathbf{H}_t}^2 + \mu_t \big(\Delta_{t+1} + (1 - \mu_t) \widetilde{\Delta}_t\big)^\top (\mathbf{H}_t \widetilde{\Delta}_t - \mathbf{g}_t).
\end{aligned}
$$

Before we proceed, we make the following assumptions, which will be addressed later.

$$
\text{Assume:} \qquad \|\mathbf{H}_t \widetilde{\Delta}_t - \mathbf{g}_t\|_{\mathbf{H}_t^{-1}} \leq \epsilon \beta \|\widetilde{\Delta}_t\|_{\mathbf{H}_t}, \qquad \mathbf{H}_t \approx_\epsilon \mathbf{H}, \tag{8}
$$

where $\epsilon = O(\frac{1}{\sqrt{d_{\text{eff}}}})$ and $\beta = \frac{\rho}{1+\rho}$ will become the convergence rate of Newton-LESS, and recall that $\rho = \frac{d_{\text{eff}}}{m - d_{\text{eff}}}$. Now, using the Cauchy-Schwartz inequality we obtain that:

$$
\begin{aligned}
\|\Delta_{t+1}\|_{\mathbf{H}_t}^2 &\leq (1 - \mu_t)^2 \|\widetilde{\Delta}_t\|_{\mathbf{H}_t}^2 + \mu_t \|\Delta_{t+1} + (1 - \mu_t) \widetilde{\Delta}_t\|_{\mathbf{H}_t} \|\mathbf{H}_t \widetilde{\Delta}_t - \mathbf{g}_t\|_{\mathbf{H}_t^{-1}} \\
&\leq (1 - \mu_t)^2 \|\widetilde{\Delta}_t\|_{\mathbf{H}_t}^2 + \epsilon \beta \mu_t \|\Delta_{t+1}\|_{\mathbf{H}_t} \|\widetilde{\Delta}_t\|_{\mathbf{H}_t} + \epsilon \beta \mu_t (1 - \mu_t) \|\widetilde{\Delta}_t\|_{\mathbf{H}_t}^2.
\end{aligned}
$$

Solving for $\|\Delta_{t+1}\|_{\mathbf{H}_t}$ (we use that if $x^2 \leq ax + b$ then $x^2 \leq a^2 + 2b$), we obtain the following:

$$
\begin{aligned}
\|\Delta_{t+1}\|_{\mathbf{H}_t}^2 &\leq 2(1 - \mu_t)^2 \|\widetilde{\Delta}_t\|_{\mathbf{H}_t}^2 + 2\epsilon \beta \mu_t (1 - \mu_t) \|\widetilde{\Delta}_t\|_{\mathbf{H}_t}^2 + \epsilon^2 \beta^2 \mu_t^2 \|\widetilde{\Delta}_t\|_{\mathbf{H}_t}^2 \\
&\leq 2\big((1 - \mu_t)^2 + \epsilon \beta \mu_t\big) \|\widetilde{\Delta}_t\|_{\mathbf{H}_t}^2.
\end{aligned}
$$

Setting $\mu_t = \frac{1}{1+\rho}$, we conclude that $\|\Delta_{t+1}\|_{\mathbf{H}_t}^2 \leq \frac{2\rho}{1+\rho} \frac{\rho+\epsilon}{1+\rho} \|\widetilde{\Delta}_t\|_{\mathbf{H}_t}^2 \leq \beta \|\widetilde{\Delta}_t\|_{\mathbf{H}_t}^2$ when $m \geq 4d_{\text{eff}} + 2$ and $\epsilon < 1/4$. Next, we return to the Newton Sketch. Recall that using Lemma 11 with the event $\mathcal{E}$ having failure probability $\delta/T$, we have:

$$
\begin{aligned}
\mathbb{E}_{\mathcal{E}} \|\widetilde{\Delta}_{t+1}\|_{\mathbf{H}_t}^2 - \|\Delta_{t+1}\|_{\mathbf{H}_t}^2 &= \rho \|\mathbf{p}_t\|_{\nabla^2 f_0(\widetilde{\mathbf{x}}_t)}^2 \pm O\big(\tfrac{\sqrt{d_{\text{eff}}}}{m}\big) \|\widetilde{\Delta}_t\|_{\mathbf{H}_t}^2 \\
&\leq \rho \|\mathbf{p}_t\|_{\mathbf{H}_t}^2 + O\big(\tfrac{\sqrt{d_{\text{eff}}}}{m}\big) \|\widetilde{\Delta}_t\|_{\mathbf{H}_t}^2, \tag{9}
\end{aligned}
$$

where we also used the fact that $\nabla^2 f_0(\widetilde{\mathbf{x}}_t) \preceq \mathbf{H}_t$. The leading term in the above decomposition can be written as follows:

$$
\begin{aligned}
\rho \|\mathbf{p}_t\|_{\mathbf{H}_t}^2 &= \rho \mu_t^2 \big(\mathbf{g}_t \widetilde{\Delta}_t - \mathbf{g}_t^\top \mathbf{H}_t^{-1} (\mathbf{H}_t \widetilde{\Delta}_t - \mathbf{g}_t)\big) \\
&= \rho \mu_t^2 \big(\widetilde{\Delta}_t^\top \mathbf{H}_t \widetilde{\Delta}_t - \widetilde{\Delta}_t^\top (\mathbf{H}_t \widetilde{\Delta}_t - \mathbf{g}_t) - \mathbf{g}_t^\top \mathbf{H}_t^{-1} (\mathbf{H}_t \widetilde{\Delta}_t - \mathbf{g}_t)\big) \\
&= \rho \mu_t^2 \|\widetilde{\Delta}_t\|_{\mathbf{H}_t}^2 - \rho \mu_t^2 (\widetilde{\Delta}_t + \mathbf{H}_t^{-1} \mathbf{g}_t)^\top (\mathbf{H}_t \widetilde{\Delta}_t - \mathbf{g}_t).
\end{aligned}
$$

Putting everything together, and then setting $\mu_t = \frac{1}{1+\rho}$, we obtain:

$$
\begin{aligned}
\mathbb{E}_{\mathcal{E}} \, \|\widetilde{\Delta}_{t+1}\|_{\mathbf{H}_t}^2 \leq{} & \Big((1-\mu_t)^2 + \rho\mu_t^2 + O\big(\tfrac{\sqrt{d_{\text{eff}}}}{m}\big)\Big)\|\widetilde{\Delta}_t\|_{\mathbf{H}_t}^2 \\
& + \Big((1+\rho)\mu_t(\Delta_{t+1} - \widetilde{\Delta}_t) + (1 - \rho\mu_t^2)\widetilde{\Delta}_t\Big)^{\top}(\mathbf{H}_t\widetilde{\Delta}_t - \mathbf{g}_t) \\
={} & \Big(\beta + O\big(\tfrac{\sqrt{d_{\text{eff}}}}{m}\big)\Big)\|\widetilde{\Delta}_t\|_{\mathbf{H}_t}^2 + \big(\Delta_{t+1} - \beta\mu_t\widetilde{\Delta}_t\big)^{\top}(\mathbf{H}_t\widetilde{\Delta}_t - \mathbf{g}_t).
\end{aligned}
$$

We can bound the second term by using Cauchy-Schwartz, the first assumption in (8) and $\mu_t \leq 1$:

$$
\Big|\big(\Delta_{t+1} - \beta\mu_t\widetilde{\Delta}_t\big)^{\top}(\mathbf{H}_t\widetilde{\Delta}_t - \mathbf{g}_t)\Big| \leq \epsilon\beta\|\Delta_{t+1}\|_{\mathbf{H}_t}\|\widetilde{\Delta}_t\|_{\mathbf{H}_t} + \epsilon\beta^2\|\widetilde{\Delta}_t\|_{\mathbf{H}_t}^2 \leq 2\epsilon\beta\|\widetilde{\Delta}_t\|_{\mathbf{H}_t}^2.
$$

Combining this with the assumption $\mathbf{H}_t \approx_\epsilon \mathbf{H}$, which implies that $\|\mathbf{v}\|_{\mathbf{H}_t}^2 \approx_\epsilon \|\mathbf{v}\|_{\mathbf{H}}^2$, we obtain:

$$
\mathbb{E}_{\delta/T} \, \frac{\|\widetilde{\Delta}_{t+1}\|_{\mathbf{H}}^2}{\|\widetilde{\Delta}_t\|_{\mathbf{H}}^2} \leq \beta \cdot \Big(1 + O\big(\tfrac{1}{\sqrt{d_{\text{eff}}}}\big)\Big). \tag{10}
$$

Note that since $\rho = \frac{d_{\text{eff}}}{m - \tilde{d}_{\text{eff}}}$, we have $\beta = \frac{d_{\text{eff}}}{m + d_{\text{eff}} - \tilde{d}_{\text{eff}}}$ and $\mu_t = \frac{d_{\text{eff}}}{m + d_{\text{eff}} - \tilde{d}_{\text{eff}}}$. Alternatively, if throughout the analysis we use $\rho = \frac{d_{\text{eff}}}{m - d_{\text{eff}}} \leq \frac{d_{\text{eff}}}{m - \tilde{d}_{\text{eff}}}$, then we obtain the simpler (and slightly weaker) convergence rate $\beta = \frac{d_{\text{eff}}}{m}$ with step size $\mu_t = 1 - \frac{d_{\text{eff}}}{m}$, as in Theorem 4.

It remains to address the assumptions from (8), and then carefully chain the expectations together. Next, we define the neighborhood $U$ in which we can establish our convergence guarantee, and show that when the iterate lies in the neighborhood, then (8) is satisfied. This part of the proof will depend on what type of function $f(\mathbf{x})$ we are minimizing.

**Lipschitz Hessian.** Suppose that function $f(\mathbf{x})$ has a Lipschitz continuous Hessian with constant $L$ (Assumption 8). We define the neighborhood $U$ through the following condition:

$$
\|\widetilde{\Delta}_t\|_{\mathbf{H}} < \frac{\sqrt{d_{\text{eff}}}}{m}\frac{(\lambda_{\min})^{3/2}}{L},
$$

where $\lambda_{\min}$ denotes the smallest eigenvalue of $\mathbf{H}$. Suppose that the condition holds for some $t$. Then, we have:

$$
\|\mathbf{H}^{-\frac{1}{2}}(\mathbf{H}_t - \mathbf{H})\mathbf{H}^{-\frac{1}{2}}\| \leq \frac{1}{\lambda_{\min}}\|\mathbf{H}_t - \mathbf{H}\| \leq \frac{L}{\lambda_{\min}}\|\widetilde{\Delta}_t\| \leq \frac{L}{(\lambda_{\min})^{3/2}}\|\widetilde{\Delta}_t\|_{\mathbf{H}} \leq \frac{\sqrt{d_{\text{eff}}}}{m} \leq \epsilon,
$$

for $\epsilon = O(\frac{1}{\sqrt{d_{\text{eff}}}})$, showing that $\mathbf{H}_t \approx_\epsilon \mathbf{H}$. In particular, this implies that $\|\mathbf{H}_t^{-1}\| \geq \frac{1}{\lambda_{\min}(1-\epsilon)}$. To get the second assumption in (8), we first follow standard analysis of the Newton's method [BV04]:

$$
\begin{aligned}
\|\mathbf{H}_t\widetilde{\Delta}_t - \mathbf{g}_t\| ={} & \left\|\mathbf{H}_t\widetilde{\Delta}_t - \Big(\int_0^1 \nabla^2 f(\mathbf{x}^* + \tau\widetilde{\Delta}_t)d\tau\Big)\widetilde{\Delta}_t\right\| \\
\leq{} & \|\widetilde{\Delta}_t\| \cdot \int_0^1 \big\|\nabla^2 f(\widetilde{\mathbf{x}}_t) - \nabla^2 f(\mathbf{x}^* + \tau\widetilde{\Delta}_t)\big\|d\tau \\
\leq{} & \|\widetilde{\Delta}_t\| \cdot \int_0^1 (1-\tau)L\|\widetilde{\Delta}_t\|d\tau \leq \frac{L}{2}\|\widetilde{\Delta}_t\|^2.
\end{aligned}
$$

Then, we simply use the fact that $\|\mathbf{H}_t^{-1}\| \geq \frac{1}{\lambda_{\min}(1-\epsilon)}$ to conclude:

$$
\begin{aligned}
\|\mathbf{H}_t\widetilde{\Delta}_t - \mathbf{g}_t\|_{\mathbf{H}_t^{-1}} \leq{} & \frac{1}{\sqrt{\lambda_{\min}(1-\epsilon)}}\|\mathbf{H}_t\widetilde{\Delta}_t - \mathbf{g}_t\| \\
\leq{} & \frac{1}{\sqrt{\lambda_{\min}(1-\epsilon)}}\frac{L}{2}\|\widetilde{\Delta}_t\|^2 \\
\leq{} & \frac{L}{(\lambda_{\min})^{3/2}}\|\widetilde{\Delta}_t\|_{\mathbf{H}}^2 \leq \epsilon\beta\|\widetilde{\Delta}_t\|_{\mathbf{H}_t},
\end{aligned}
$$

since $\beta = O(\frac{d_{\text{eff}}}{m})$, thus establishing the assumptions from (8).

**Self-concordant function.** Suppose that function $f(\mathbf{x})$ is self-concordant (Assumption 9). We will define the neighborhood $U$ through the following condition:

$$\|\widetilde{\Delta}_t\|_{\mathbf{H}} < \frac{\sqrt{d_{\text{eff}}}}{m}.$$

Now, using a standard property of self-concordant functions [BV04, Chapter 9], we have:

$$(1 - \|\widetilde{\Delta}_t\|_{\mathbf{H}})^2 \, \mathbf{H} \preceq \mathbf{H}_t \preceq \frac{1}{(1 - \|\widetilde{\Delta}_t\|_{\mathbf{H}})^2} \, \mathbf{H},$$

and note that $\frac{1}{(1-\|\widetilde{\Delta}_t\|_{\mathbf{H}})^2} \leq 1 + \epsilon$ for $\epsilon = O(\frac{1}{\sqrt{d_{\text{eff}}}})$, so it follows that $\mathbf{H}_t \approx_\epsilon \mathbf{H}$. Furthermore, for self-concordant functions, it follows that:

$$
\begin{aligned}
\|\mathbf{H}_t \widetilde{\Delta}_t - \mathbf{g}_t\|_{\mathbf{H}_t^{-1}} &= \left\| \mathbf{H}_t^{\frac{1}{2}} \widetilde{\Delta}_t - \mathbf{H}_t^{-\frac{1}{2}} \Big( \int_0^1 \nabla^2 f(\mathbf{x}^* + \tau \widetilde{\Delta}_t) d\tau \Big) \widetilde{\Delta}_t \right\| \\
&= \left\| \Big( \int_0^1 \big( \mathbf{I} - \mathbf{H}_t^{-\frac{1}{2}} \nabla^2 f(\mathbf{x}^* + \tau \widetilde{\Delta}_t) \mathbf{H}_t^{-\frac{1}{2}} \big) d\tau \Big) \mathbf{H}_t^{\frac{1}{2}} \widetilde{\Delta}_t \right\| \\
&\leq \|\widetilde{\Delta}_t\|_{\mathbf{H}_t} \cdot \int_0^1 \big\| \mathbf{I} - \mathbf{H}_t^{-\frac{1}{2}} \nabla^2 f(\mathbf{x}^* + \tau \widetilde{\Delta}_t) \mathbf{H}_t^{-\frac{1}{2}} \big\| d\tau \\
&\leq \|\widetilde{\Delta}_t\|_{\mathbf{H}_t} \cdot \int_0^1 \frac{1}{(1 - \tau\|\widetilde{\Delta}_t\|_{\mathbf{H}_t})^2} d\tau = \frac{\|\widetilde{\Delta}_t\|_{\mathbf{H}_t}^2}{1 - \|\widetilde{\Delta}_t\|_{\mathbf{H}_t}}.
\end{aligned}
$$

Using the neighborhood condition, we conclude that $\frac{\|\widetilde{\Delta}_t\|_{\mathbf{H}_t}^2}{1 - \|\widetilde{\Delta}_t\|_{\mathbf{H}_t}} \leq O(\frac{\sqrt{d_{\text{eff}}}}{m})\|\widetilde{\Delta}_t\|_{\mathbf{H}_t} \leq \epsilon\beta\|\widetilde{\Delta}_t\|_{\mathbf{H}_t}$.

**Chaining the expectations.** Let $\mathcal{E}_t$ denote the high-probability event corresponding to the conditional expectation in (10) for the iteration $t$. It remains to show that after conditioning on event $\mathcal{E} = \bigwedge_{t=0}^{T-1} \mathcal{E}_t$, we maintain that $\widetilde{\mathbf{x}}_t \in U$ for all $t$. Assume that this holds for $t = 0$. Then, it suffices to show that $\|\widetilde{\Delta}_{t+1}\|_{\mathbf{H}} \leq \|\widetilde{\Delta}_t\|_{\mathbf{H}}$ for every $t$ almost surely (conditioned on $\mathcal{E}$). Recall that Theorem 6 implies that conditioned on $\mathcal{E}_t$ we have

$$\|\mathbf{I} - \widetilde{\mathbf{Q}}\| \leq \eta,$$

where Lemma 7 ensures that $\eta$ is small. We use this to show the following coarse convergence guarantee that holds almost surely conditioned on $\mathcal{E}_t$, but is substantially weaker than $\beta$. First, note that using the derivation as in the proof of Lemma 11 and the analysis of the exact Newton step,

$$
\begin{aligned}
\|\widetilde{\Delta}_{t+1}\|_{\mathbf{H}_t}^2 &\leq \|\Delta_{t+1}\|_{\mathbf{H}_t}^2 + O\big(\|\mathbf{I} - \widetilde{\mathbf{Q}}\|\big) \cdot \|\widetilde{\Delta}_t\|_{\mathbf{H}_t}^2 \\
&\leq \big(\beta + O(\eta)\big) \cdot \|\widetilde{\Delta}_t\|_{\mathbf{H}_t}^2.
\end{aligned}
$$

Using a sufficiently large constant $C$ in Lemma 7 so that $\beta + O(\eta)$ is small enough, and given the assumption $\mathbf{H}_t \approx_\epsilon \mathbf{H}$, we obtain $\|\widetilde{\Delta}_{t+1}\|_{\mathbf{H}}^2 \leq \|\widetilde{\Delta}_t\|_{\mathbf{H}}^2$. Thus, we conclude that all of the iterates will lie in the neighborhood $U$, and so (10) will hold for all $t = 0, 1, ..., T-1$. Finally, note that by the union bound, event $\mathcal{E}$ holds with probability $1 - \delta$, which completes the proof.

**Lower-bound from Theorem 1** The matching lower-bound from Theorem 1 holds only in the unregularized setting. In this case, we have $\nabla^2 f_0(\widetilde{\mathbf{x}}_t) = \mathbf{H}_t$, so instead of an inequality in (9), we can obtain a two-sided approximation. The rest of the proof proceeds identically.

## C  Sketches Satisfying Structural Conditions (Proof of Lemma 7)

In this section, we prove Lemma 7, showing that sub-Gaussian, LESS, and LESS-uniform embeddings all satisfy the assumptions of Theorem 6, which are derived from Conditions 1 and 2. This analysis follows along similar lines as in [DLDM21], except for extending LESS embeddings to LESS-uniform, and allowing for the presence of regularization.

## C.1 Sub-Gaussian embeddings

For sub-Gaussian embeddings, both conditions follow from existing results. To establish Condition 1, we rely on a covariance estimation result of [KL17], stated as in [MZ20]. Here, we say that a random vector $\mathbf{x}$ is sub-Gaussian if $\mathbf{v}^\top \mathbf{x}$ is a sub-Gaussian variable for all unit vectors $\mathbf{v}$.

**Lemma 15** ([MZ20, Theorem 1.4]). *For $i \in \{1, \ldots, m\}$, let $\mathbf{x}_i \in \mathbb{R}^n$ be independent sub-Gaussian random vectors such that $\mathbb{E}[\mathbf{x}_i] = 0$ and $\mathbb{E}[\mathbf{x}_i \mathbf{x}_i^\top] = \boldsymbol{\Sigma}$. Then, it holds with probability at least $1 - 2\exp(-t^2)$ that*

$$\left\| \frac{1}{m} \sum_{i=1}^m \mathbf{x}_i \mathbf{x}_i^\top - \boldsymbol{\Sigma} \right\| \leq C \|\boldsymbol{\Sigma}\| \left( \sqrt{\frac{\operatorname{tr}\boldsymbol{\Sigma}/\|\boldsymbol{\Sigma}\|}{m}} + \frac{\operatorname{tr}\boldsymbol{\Sigma}/\|\boldsymbol{\Sigma}\|}{m} + \frac{t}{\sqrt{m}} + \frac{t^2}{m} \right).$$

Now, to establish Condition 1 for the sub-Gaussian sketching matrix $\mathbf{S}$, i.e., where the $m$ rows are distributed as $\frac{1}{\sqrt{m-d_{\mathrm{eff}}}} \mathbf{s}_i^\top$ for $\mathbf{s}_i$ having i.i.d. zero mean, unit variance and sub-Gaussian entries, we let $\mathbf{x}_i = \mathbf{U}^\top \mathbf{s}_i$. Recall that $\mathbf{U} = \mathbf{A}\mathbf{H}^{-\frac{1}{2}}$ for $\mathbf{H} = \mathbf{A}^\top \mathbf{A} + \mathbf{C}$, so $\|\mathbf{U}\| \leq 1$, so it follows that $\mathbf{x}_i$ is a sub-Gaussian random vector. Therefore, letting $\gamma = \frac{m}{m-d_{\mathrm{eff}}}$ and $\boldsymbol{\Sigma} = \mathbf{U}^\top \mathbf{U}$, we have $\mathbb{E}[\mathbf{x}_i \mathbf{x}_i^\top] = \boldsymbol{\Sigma}$ and with probability $1 - \delta$:

$$\|\mathbf{U}^\top \mathbf{S}^\top \mathbf{S} \mathbf{U} - \boldsymbol{\Sigma}\| \leq \gamma \cdot \left\| \frac{1}{m} \sum_{i=1}^m \mathbf{x}_i \mathbf{x}_i^\top - \boldsymbol{\Sigma} \right\| + (\gamma - 1) \cdot \|\boldsymbol{\Sigma}\|$$

$$\leq C \left( \sqrt{\frac{d_{\mathrm{eff}}}{m}} + \sqrt{\frac{\log(1/\delta)}{m}} \right) + \frac{d_{\mathrm{eff}}}{m - d_{\mathrm{eff}}},$$

thus setting $m \geq O(1) \cdot (d_{\mathrm{eff}} + \log(1/\delta))/\eta^2$, we can bound the above by $\eta$, obtaining Condition 1.

To show Condition 2 for sub-Gaussian embeddings, we can again rely on a more general moment bound for quadratic forms, which is a special case of Lemma B.26 in [BS10].

**Lemma 16** ([BS10]). *Let $\mathbf{M}$ be a $n \times n$ matrix, and let $\mathbf{x}$ be an $n$-dimensional random vector with independent, mean zero, unit variance entries such that $\mathbb{E}[x_i^4] = O(1)$. Then,*

$$\operatorname{Var}[\mathbf{x}^\top \mathbf{M} \mathbf{x}] \leq O(1) \cdot \operatorname{tr}(\mathbf{M}\mathbf{M}^\top).$$

To obtain Condition 2, we simply set $\mathbf{x} = \mathbf{s}$ and $\mathbf{M} = \mathbf{U}\mathbf{B}\mathbf{U}^\top$. Note that since $\|\mathbf{U}\| \leq 1$, we have $\operatorname{tr}(\mathbf{M}\mathbf{M}^\top) = \operatorname{tr}(\mathbf{U}\mathbf{B}\mathbf{U}^\top \mathbf{U}\mathbf{B}\mathbf{U}^\top) \leq \operatorname{tr}(\mathbf{U}\mathbf{B}^2 \mathbf{U}^\top)$.

## C.2 LESS embeddings: Condtion 1

Now, we demonstrate that Condition 1 also holds for LESS embeddings. We will use the following matrix concentration inequality which is a straightforward combination of two standard results.

**Lemma 17** ([Tro12, Theorem 6.2] and [Tro15, Theorem 7.7.1]). *For $i = 1, 2, \ldots$, consider a finite sequence $\mathbf{X}_i$ of $d \times d$ independent symmetric random matrices such that $\mathbb{E}[\mathbf{X}_i] = \mathbf{0}$, and one of the following holds for all $i$:*

1. $\mathbb{E}[\mathbf{X}_i^p] \preceq \frac{p!}{2} \cdot R^{p-2} \mathbf{A}_i^2$ *for* $p = 2, 3, \ldots$;

2. $\|\mathbf{X}_i\| \leq R$ *and* $\mathbb{E}[\mathbf{X}_i^2] \preceq \mathbf{A}_i^2$.

*Then, defining the variance matrix $\mathbf{V} = \sum_i \mathbf{A}_i^2$, parameter $\sigma^2 = \|\mathbf{V}\|$ and $d_{\mathrm{eff}} = \operatorname{tr}(\mathbf{V})/\|\mathbf{V}\|$, for any $t \geq \sigma + R$ we have:*

$$\Pr \left\{ \lambda_{\max} \left( \sum_i \mathbf{X}_i \right) \geq t \right\} \leq 4 d_{\mathrm{eff}} \cdot \exp \left( \frac{-t^2/2}{\sigma^2 + Rt} \right).$$

Before we can use matrix concentration, we must first establish high-probability concentration of the quadratic form $\mathbf{s}^\top \mathbf{U}\mathbf{U}^\top \mathbf{s}$, for a leverage score sparsified sub-Gaussian random vector $\mathbf{s}$. This is an analog of the Hanson-Wright inequality, which holds for non-sparsified sub-Gaussian random vectors, as given below.

**Lemma 18** (Hanson-Wright inequality, [RV13, Theorem 1.1]). *Let $\mathbf{x}$ have independent sub-Gaussian entries with mean zero and unit variance. Then, there is $c = \Omega(1)$ such that for any $n \times n$ matrix $\mathbf{B}$ and $t \geq 0$,*

$$\Pr\left\{|\mathbf{x}^\top \mathbf{B}\mathbf{x} - \operatorname{tr}(\mathbf{B})| \geq t\right\} \leq 2\exp\left(-c\min\left\{\frac{t^2}{\|\mathbf{B}\|_F^2}, \frac{t}{\|\mathbf{B}\|}\right\}\right).$$

Our version of this result for sparsified sub-Gaussian vectors is an extension of Lemma 31 of [DLDM21], introducing the effective dimension $d_{\text{eff}}$ as opposed to the regular dimension $d$, and allowing a broader class of sparsifiers, so that we can cover the results for LESS-uniform embeddings.

**Lemma 19.** *Let $\mathbf{U} = \mathbf{A}\mathbf{H}^{-\frac{1}{2}}$ for $\mathbf{H} = \mathbf{A}^\top \mathbf{A} + \mathbf{C}$, and let $d_{\text{eff}} = \operatorname{tr}(\mathbf{U}^\top \mathbf{U})$. Let $\boldsymbol{\xi}$ be a $(p, s)$-sparsifier and $\mathbf{x}$ have indepedent sub-Gaussian entries with mean zero and unit variance. If $p_i = \Omega(l_i(\mathbf{A}, \mathbf{C})/s)$ for all $i$, then for any $t \geq Cd_{\text{eff}}$, vector $\mathbf{s} = \mathbf{x} \circ \boldsymbol{\xi}$ satisfies:*

$$\Pr\left\{\mathbf{s}^\top \mathbf{U}\mathbf{U}^\top \mathbf{s} \geq t\right\} \leq \exp\left(-c\left(\sqrt{t} + t/d_{\text{eff}}\right)\right).$$

*Proof.* The analysis follows along the same lines as the proof of Lemma 31 in [DLDM21]. First, we define the shorthand $\bar{\mathbf{U}} = \operatorname{diag}(\boldsymbol{\xi})\mathbf{U}$, and use Lemma 17 to bound the spectral norm $\|\bar{\mathbf{U}}\|$. Observe that from the definition of the sparsifier $\boldsymbol{\xi}$ we have the following decomposition: $\bar{\mathbf{U}}^\top \bar{\mathbf{U}} = \sum_{i=1}^s \frac{1}{sp_{t_i}} \mathbf{u}_{t_i}\mathbf{u}_{t_i}^\top$, where $\mathbf{u}_i^\top$ denotes the $i$th row of $\mathbf{U}$ and $t_1, ..., t_s$ are the independently sampled indices from $p$. Note that since $l_i(\mathbf{A}, \mathbf{C}) = \|\mathbf{u}_i\|^2$, we have that $p_i = \Omega(\|\mathbf{u}_i\|^2/s)$, so $\mathbf{X}_i = \frac{1}{sp_{t_i}}\mathbf{u}_{t_i}\mathbf{u}_{t_i}^\top - \frac{1}{s}\mathbf{U}^\top \mathbf{U}$ satisfies $\mathbb{E}[\mathbf{X}_i] = \mathbf{0}$, $\|\mathbf{X}_i\| = O(1)$, and $\mathbb{E}[\mathbf{X}_i^2] = O(1/s) \cdot \mathbf{I}$. So, using Lemma 17 with $\sigma^2 = R = O(1)$, for any $t \geq Cd_{\text{eff}}$ we have $\Pr\{\|\bar{\mathbf{U}}\|^2 \geq \sqrt{t}\} \leq \exp(-c\sqrt{t})$, with $c = \Omega(1)$. Using the fact that $\|\bar{\mathbf{U}}\|^2 \leq \operatorname{tr}(\bar{\mathbf{U}}^\top \bar{\mathbf{U}}) \leq Cd_{\text{eff}}$ almost surely, it follows that the event $\mathcal{E} : \|\bar{\mathbf{U}}\|^2 \leq \min\{\sqrt{t}, Cd_{\text{eff}}\}$ has probability $1 - \exp(-c(\sqrt{t} + t/d_{\text{eff}}))$. Now, it suffces to condition on $\boldsymbol{\xi}$ and apply the Hanson-Wright inequality (Lemma 18), concluding that:

$$\Pr\left\{\mathbf{x}^\top \bar{\mathbf{U}}\bar{\mathbf{U}}^\top \mathbf{x} \geq Cd_{\text{eff}} + t \mid \boldsymbol{\xi}, \mathcal{E}\right\} \leq 2\exp\left(-c\min\left\{\frac{t^2}{\|\bar{\mathbf{U}}\bar{\mathbf{U}}\|_F^2}, \frac{t}{\|\bar{\mathbf{U}}\bar{\mathbf{U}}^\top\|}\right\}\right)$$
$$\leq 2\exp(-\Omega(\sqrt{t} + t/d_{\text{eff}})),$$

which completes the proof. ∎

By appropriately integrating out the concentration inequality from Lemma 19, as in Lemma 30 of [DLDM21] but replacing $d$ with $d_{\text{eff}}$, we can show the following matrix moment bound.

**Lemma 20.** *Under the assumptions of Lemma 19, for all $p = 2, 3, ...$ we have:*

$$\left\|\mathbb{E}\left[\left(\mathbf{U}^\top \mathbf{s}\mathbf{s}^\top \mathbf{U} - \mathbf{U}^\top \mathbf{U}\right)^p\right]\right\| \leq \frac{p!}{2} \cdot (Cd_{\text{eff}})^{p-1}.$$

*Proof.* First, we bound the expression in terms of the quadratic form $\mathbf{s}^\top \mathbf{U}\mathbf{U}^\top \mathbf{s}$, so that we can use the concentration inequality from Lemma 19. To that end, we have:

$$\mathbb{E}\left[\left(\mathbf{U}^\top \mathbf{s}\mathbf{s}^\top \mathbf{U} - \mathbf{U}^\top \mathbf{U}\right)^p\right] \preceq \mathbb{E}\left[\left\|\mathbf{U}^\top \mathbf{s}\mathbf{s}^\top \mathbf{U} - \mathbf{U}^\top \mathbf{U}\right\|^{p-2}\left(\mathbf{U}^\top \mathbf{s}\mathbf{s}^\top \mathbf{U} - \mathbf{U}^\top \mathbf{U}\right)^2\right]$$

$$\overset{(*)}{\preceq} \mathbb{E}\left[\left(\mathbf{s}^\top \mathbf{U}\mathbf{U}^\top \mathbf{s} + d_{\text{eff}}\right)^{p-2}\left(2(\mathbf{U}^\top \mathbf{s}\mathbf{s}^\top \mathbf{U})^2 + 2(\mathbf{U}^\top \mathbf{U})^2\right)\right]$$

$$\preceq 2\mathbb{E}\left[\left(\mathbf{s}^\top \mathbf{U}\mathbf{U}^\top \mathbf{s} + d_{\text{eff}}\right)^{p-1}\mathbf{U}^\top \mathbf{s}\mathbf{s}^\top \mathbf{U}\right] + 2\mathbb{E}\left[\left(\mathbf{s}^\top \mathbf{U}\mathbf{U}^\top \mathbf{s} + d_{\text{eff}}\right)^{p-2}\right] \cdot \mathbf{I},$$

where in $(*)$ we used the fact that function $f(x) = x^2$ is operator convex. Now, integrating out the concentration inequality from Lemma 19 for each of the two terms (following the steps of [DLDM21, Appendix D.2]), we obtain the desired bound. ∎

We can now apply Lemma 17 with $\mathbf{X}_i = \frac{1}{m}\mathbf{U}^\top\mathbf{s}_i\mathbf{s}_i^\top\mathbf{U} - \frac{1}{m}\mathbf{U}^\top\mathbf{U}$, and $\sigma^2 = R = O(d_{\text{eff}}/m)$, obtaining that:

$$\Pr\left\{\|\gamma^{-1}\mathbf{U}^\top\mathbf{S}^\top\mathbf{S}\mathbf{U} - \mathbf{U}^\top\mathbf{U}\| \geq \eta\right\} \leq d_{\text{eff}} \cdot \exp\left(-\Omega(\eta^2 m/d_{\text{eff}})\right).$$

Setting $m \geq Cd_{\text{eff}}\log(d_{\text{eff}}/\delta)/\eta^2$, we obtain the desired bound. Note that we must account again for the scaling $\gamma = \frac{m}{m-d_{\text{eff}}}$, which gets absorbed into the error $\eta$.

Finally, observe that the conditions imposed on the sparsifier $\boldsymbol{\xi}$ in Lemma 19 encompass both LESS and LESS-uniform embeddings. In the case of LESS, we can simply let $s \approx_{1/2} d_{\text{eff}}$, and then the condition on sparsifying distribution is $p_i = \Omega(l_i(\mathbf{A},\mathbf{C})/d_{\text{eff}})$. On the other hand, for LESS-uniform, as long as $s = \Omega(\tau d_{\text{eff}})$ where $\tau = \frac{n}{d_{\text{eff}}}\max_i l_i(\mathbf{A},\mathbf{C})$ is the coherence of $\mathbf{A}$, it follows that $\frac{1}{n} = \Omega(l_i(\mathbf{A},\mathbf{C})/s)$ for all $i$'s, so a uniformly sparsifying distribution suffices.

### C.3 LESS embeddings: Condition 2

Here, we prove a result that is similar to the so-called Restricted Bai-Silverstein inequality from [DLDM21, Lemma 28]. Our assumptions on the sparsifier are somewhat weaker, to account for LESS-uniform embeddings and for the presence of regularization.

**Lemma 21.** *Let* $\mathbf{U} = \mathbf{A}\mathbf{H}^{-\frac{1}{2}}$ *for* $\mathbf{H} = \mathbf{A}^\top\mathbf{A} + \mathbf{C}$. *Let* $\boldsymbol{\xi}$ *be a* $(p,s)$-*sparsifier and* $\mathbf{x}$ *have indepedent sub-Gaussian entries with mean zero and unit variance. If* $p_i = \Omega(l_i(\mathbf{A},\mathbf{C})/s)$ *for all* $i$, *then for all* $d \times d$ *psd matrices* $\mathbf{B}$, *vector* $\mathbf{s} = \mathbf{x} \circ \boldsymbol{\xi}$ *satisfies:*

$$\text{Var}[\mathbf{s}^\top\mathbf{U}\mathbf{B}\mathbf{U}^\top\mathbf{s}] \leq O(1) \cdot \text{tr}(\mathbf{U}\mathbf{B}^2\mathbf{U}^\top).$$

*Proof.* Let $\bar{\mathbf{U}} = \text{diag}(\boldsymbol{\xi})\mathbf{U}$. We start with a decomposition of the variance:

$$\text{Var}[\mathbf{s}^\top\mathbf{U}\mathbf{B}\mathbf{U}^\top\mathbf{s}] = \mathbb{E}\left[(\mathbf{x}^\top\bar{\mathbf{U}}\mathbf{B}\bar{\mathbf{U}}^\top\mathbf{x} - \text{tr}(\bar{\mathbf{U}}\mathbf{B}\bar{\mathbf{U}}^\top) + \text{tr}(\bar{\mathbf{U}}\mathbf{B}\bar{\mathbf{U}}^\top) - \text{tr}(\mathbf{B}))^2\right]$$
$$= \mathbb{E}\left[\text{Var}[\mathbf{x}^\top\bar{\mathbf{U}}\mathbf{B}\bar{\mathbf{U}}^\top\mathbf{x} \mid \bar{\mathbf{U}}]\right] + \text{Var}[\text{tr}(\bar{\mathbf{U}}\mathbf{B}\bar{\mathbf{U}}^\top)].$$

Recall that $\bar{\mathbf{U}}^\top\bar{\mathbf{U}} = \sum_{i=1}^s \frac{1}{sp_{t_i}}\mathbf{u}_{t_i}\mathbf{u}_{t_i}^\top$, where $\mathbf{u}_i^\top$ is the $i$th row of $\mathbf{U}$ and $p_i = \Omega(\|\mathbf{u}_i\|^2/s)$. Then

$$\text{Var}\left[\text{tr}(\bar{\mathbf{U}}\mathbf{B}\bar{\mathbf{U}}^\top)\right] = s\,\text{Var}\left[\frac{\mathbf{u}_{t_1}^\top\mathbf{B}\mathbf{u}_{t_1}}{sp_{t_1}}\right] \leq \mathbb{E}\left[\frac{\|\mathbf{u}_{t_1}\|^2}{sp_{t_1}}\frac{\mathbf{u}_{t_1}^\top\mathbf{B}^2\mathbf{u}_{t_1}}{p_{t_1}}\right] = O(1)\,\text{tr}(\mathbf{U}\mathbf{B}^2\mathbf{U}^\top),$$

where we use that $\mathbf{U}^\top\mathbf{U} \preceq \mathbf{I}$. Next, we use the classical Bai-Silverstein inequality (Lemma 16):

$$\mathbb{E}\left[\text{Var}[\mathbf{x}^\top\bar{\mathbf{U}}\mathbf{B}\bar{\mathbf{U}}^\top\mathbf{x} \mid \bar{\mathbf{U}}]\right] \leq O(1) \cdot \mathbb{E}\left[\text{tr}((\bar{\mathbf{U}}\mathbf{B}\bar{\mathbf{U}}^\top)^2)\right] = O(1) \cdot \mathbb{E}\left[\text{tr}\left(\left(\sum_{i=1}^s \frac{1}{sp_{t_i}}\mathbf{B}\mathbf{u}_{t_i}\mathbf{u}_{t_i}^\top\right)^2\right)\right]$$
$$= O(1)\,\text{tr}(\mathbf{U}\mathbf{B}^2\mathbf{U}^\top) + O(1)\,\text{tr}((\mathbf{U}\mathbf{B}\mathbf{U}^\top)^2),$$

where the last step follows by breaking down the expanded square into the diagonal part and the cross-terms. Since $\text{tr}((\mathbf{U}\mathbf{B}\mathbf{U}^\top)^2) \leq \text{tr}(\mathbf{U}\mathbf{B}^2\mathbf{U}^\top)$, this completes the proof. ∎

## D Distributed Averaging for Newton-LESS

An important property of the Gaussian Newton Sketch is that it produces unbiased estimates of the exact Newton step. This is useful in distributed settings, where we can construct multiple independent estimates in parallel, and then produce an improved estimate by averaging them together. Newton-LESS retains this unbiasedness property, up to a small error, which also makes it amenable to distributed averaging. This near-unbiasedness of LESS embeddings follows from the characterization of the first inverse moment of the sketched Hessian (see [DLDM21] and the first part of Theorem 6).

In this section, we show that the near-unbiasedness of LESS embeddings can be combined with our new convergence analysis to provide improved convergence rates for Distributed Newton-LESS:

$$\widetilde{\mathbf{x}}_{t+1} = \widetilde{\mathbf{x}}_t - \frac{\mu_t}{q}\sum_{i=1}^q \left(\mathbf{A}_{f_0}(\widetilde{\mathbf{x}}_t)^\top\mathbf{S}_{t,i}^\top\mathbf{S}_{t,i}\mathbf{A}_{f_0}(\widetilde{\mathbf{x}}_t) + \nabla^2 g(\widetilde{\mathbf{x}}_t)\right)^{-1}\nabla f(\widetilde{\mathbf{x}}_t), \tag{11}$$

where $\mathbf{S}_{t,i}$ are independently drawn LESS embedding matrices. To adapt our analysis for this algorithm, we extend the characterization from Lemma 11.

**Lemma 22.** *Fix* $\mathbf{H}_t = \nabla^2 f(\widetilde{\mathbf{x}}_t)$ *and let* $\widetilde{\mathbf{x}}_{t+1}$ *be as in* (11) *with* $\mathbf{S}_{t,i}$ *as in Lemma* 7 *(i.e., sub-Gaussian, LESS or LESS-uniform). Also, suppose that the exact Newton step* $\mathbf{x}_{t+1} = \widetilde{\mathbf{x}}_t - \mu_t \mathbf{H}_t^{-1}\mathbf{g}_t$ *is a descent direction, i.e.,* $\|\Delta_{t+1}\|_{\mathbf{H}_t} \leq \|\widetilde{\Delta}_t\|_{\mathbf{H}_t}$ *where* $\Delta_{t+1} = \mathbf{x}_{t+1} - \mathbf{x}^*$ *and* $\widetilde{\Delta}_t = \widehat{\mathbf{x}}_t - \mathbf{x}^*$. *Then, letting* $\rho = \frac{d_{\mathrm{eff}}(\widetilde{\mathbf{x}}_t)}{m - \tilde{d}_{\mathrm{eff}}(\widetilde{\mathbf{x}}_t)}$, *we have:*

$$\mathbb{E}_{q\delta}\|\widetilde{\Delta}_{t+1}\|_{\mathbf{H}_t}^2 = \|\Delta_{t+1}\|_{\mathbf{H}_t}^2 + \tfrac{\rho}{q}\|\mathbf{x}_{t+1} - \widetilde{\mathbf{x}}_t\|_{\nabla^2 f_0(\widetilde{\mathbf{x}}_t)}^2 \pm O\big(\tfrac{\sqrt{d_{\mathrm{eff}}}}{m}\big)\|\widetilde{\Delta}_t\|_{\mathbf{H}_t}^2.$$

*Proof.* The proof is analogous to the proof of Lemma 11, except we must replace $\widetilde{\mathbf{Q}}$ with

$$\bar{\mathbf{Q}} = \frac{1}{q}\sum_{i=1}^q \widetilde{\mathbf{Q}}_i, \qquad \text{for} \qquad \widetilde{\mathbf{Q}}_i = \mathbf{H}_t^{\frac{1}{2}}(\mathbf{A}_{f_0}(\widetilde{\mathbf{x}}_t)^\top \mathbf{S}_{t,i}^\top \mathbf{S}_{t,i}\mathbf{A}_{f_0}(\widetilde{\mathbf{x}}_t) + \nabla^2 g(\widetilde{\mathbf{x}}_t))^{-1}\mathbf{H}_t^{\frac{1}{2}}.$$

Each $\widetilde{\mathbf{Q}}_i$ satisfies the first and second moment characterizations from Theorem 6. Let $\mathcal{E} = \bigwedge_{i=1}^q \mathcal{E}_i$ denote the intersection of the corresponding $1-\delta$ probability events. Then, $\|\mathbb{E}_{\mathcal{E}}[\bar{\mathbf{Q}}] - \mathbf{I}\| \leq O\big(\tfrac{\sqrt{d_{\mathrm{eff}}}}{m}\big)$ and also:

$$\mathbb{E}_{\mathcal{E}}[\bar{\mathbf{Q}}^2] - \mathbf{I} = \frac{1}{q^2}\sum_{i=1}^q \mathbb{E}_{\mathcal{E}}[\widetilde{\mathbf{Q}}_i^2] - \sum_{i \neq j}\mathbb{E}_{\mathcal{E}}[\widetilde{\mathbf{Q}}_i]\mathbb{E}_{\mathcal{E}}[\widetilde{\mathbf{Q}}_j] - \mathbf{I}$$

$$= \frac{1}{q}\big(\mathbb{E}_{\mathcal{E}}[\widetilde{\mathbf{Q}}_1^2] - \mathbf{I}\big) + \frac{q(q-1)}{q^2}\big(\mathbb{E}_{\mathcal{E}}[\widetilde{\mathbf{Q}}_1]^2 - \mathbf{I}\big),$$

so using that $\|\mathbb{E}_{\mathcal{E}}[\widetilde{\mathbf{Q}}_1^2] - (\mathbf{I} + \rho\,\mathbf{U}^\top\mathbf{U})\| \leq O\big(\tfrac{\sqrt{d_{\mathrm{eff}}}}{m}\big)$, where $\mathbf{U} = \mathbf{A}_{f_0}(\widetilde{\mathbf{x}})\mathbf{H}_t^{-\frac{1}{2}}$, we get:

$$\|\mathbb{E}_{\mathcal{E}}[\bar{\mathbf{Q}}^2] - (\mathbf{I} + \tfrac{\rho}{q}\mathbf{U}^\top\mathbf{U})\| \leq O\big(\tfrac{\sqrt{d_{\mathrm{eff}}}}{m}\big).$$

The rest of the proof follows identically as in Lemma 11, using $\bar{\mathbf{Q}}$ in place of $\widetilde{\mathbf{Q}}$. Note that, using the union bound, we can show that the probability of $\mathcal{E}$ is at least $1 - q\delta$. ∎

From this lemma, repeating the local convergence analysis of Theorem 10, we obtain that in the neighborhood of $\mathbf{x}^*$, setting $\mu_t = \frac{q(m - \tilde{d}_{\mathrm{eff}})}{d_{\mathrm{eff}} + q(m - \tilde{d}_{\mathrm{eff}})}$, Distributed Newton-LESS achieves:

$$\left(\mathbb{E}_{qT\delta}\frac{\|\widetilde{\mathbf{x}}_T - \mathbf{x}^*\|_{\mathbf{H}}^2}{\|\widetilde{\mathbf{x}}_0 - \mathbf{x}^*\|_{\mathbf{H}}^2}\right)^{1/T} \leq \frac{d_{\mathrm{eff}}}{d_{\mathrm{eff}} + q(m - \tilde{d}_{\mathrm{eff}})} + O\Big(\frac{\sqrt{d_{\mathrm{eff}}}}{m}\Big),$$

and for the unregularized case, where $d_{\mathrm{eff}} = \tilde{d}_{\mathrm{eff}} = d$, we can obtain a matching lower bound on the convergence rate. This shows that the convergence rate of Newton-LESS can be substantially improved via distributed averaging.

# E   Additional Numerical Experiments and Implemention Details

Experiments are implemented in Python using the Pytorch module on Amazon Sagemaker instances with CPUs with 256 gigabytes of memory and GPUs NVIDIA Tesla V100. The code is publicly available at https://github.com/lessketching/newtonsketch.

## E.1   Sketching matrices

Given a data matrix $\mathbf{A} \in \mathbb{R}^{n \times d}$, we follow the procedure described in [DMIMW12b] (see Algorithm 1 therein) for fast approximation of the leverage scores. We use these approximate leverage scores to compute the RSS-lev-score embedding.

For LESS embeddings, we report the performance of the computationally most efficient method between using the approximate leverage scores, or, pre-processing the data matrix $\mathbf{A}$ by a Hadamard matrix $\mathbf{H}$ and then using a uniformly sparsified sketching matrix. Preprocessing with a Hadamard matrix uniformizes the leverage scores, so this second option is a valid implementation of a LESS embedding (see [DLDM21] for a detailed discussion). We found this second option to be the fastest method in practice.

For a LESS-uniform embedding $\mathbf{S} \in \mathbb{R}^{m \times n}$, we fix a number of non-zero entries per row to be $s$. For each row $\mathbf{s}_i^\top$, we sample $s$ indices $\{i_1, \ldots, i_s\}$ in $\{1, \ldots, n\}$ uniformly at random with replacement. Each entry $S_{ii_j}$ is then chosen uniformly at random in $\{\pm(n/ms)^{1/2}\}$. We choose to sample with replacement for maximal computational efficiency. In our experiments, the number of non-zero entries $s$ is small in comparison to the sample size $n$, so the probability of sampling twice the same index remains very small.

## E.2 Datasets

The high-coherence synthetic data matrix $\mathbf{A}$ is generated as follows. We construct a covariance matrix $\boldsymbol{\Sigma} \in \mathbb{R}^{d \times d}$ with entries $\boldsymbol{\Sigma}_{ij} = 2 \cdot 0.5^{|i-j|}$. The rows $\mathbf{a}_i$ of $\mathbf{A}$ are then sampled independently as $\mathbf{a}_i \sim \mathbf{g}_i/\sqrt{z_i}$ where $\mathbf{g}_i \sim \mathcal{N}(0, \Sigma)$ and $z_i$ follows a Gamma distribution with shape $1/2$ and scale 2. We use $n = 16384$ and $d = 256$.

We downloaded the Musk and CIFAR-10 datasets from https://www.openml.org/. The Musk data matrix has size $n = 4096$ and $d = 256$. The sample size of the CIFAR-10 dataset is $n = 50000$. We transform each image using a random features map that approximates the Gaussian kernel $\exp(-\gamma x^2)$ with bandwith $\gamma = 0.02$, and we use $d = 2000$ random cosine components. We partition the ten classes of CIFAR-10 into two groups with labels 0 and 1.

For the WESAD dataset [SRD$^+$18], we used the data obtained from the E4 Empatica device and we filtered the data over windows of one second.[2] This results in a sample size $n = 262144$. Then we applied a random features map that approximates the Gaussian kernel $\exp(-\gamma x^2)$ with $\gamma = 0.01$ and we use $d = 2000$ components.

## E.3 Least squares regression

We first consider least squares regression. On Figure 4, we report the relative error versus number of iterations, as well as the relative error versus wall-clock time for the Newton Sketch. We compare to Gaussian embeddings, the SRHT, uniformly random row sampling matrices (RRS) and random row sampling based on approximate leverage scores (RRS-lev-scores). As predicted by our theory, LESS embeddings have convergence rate scaling as $d/m$. This is similar to the convergence rate of the Newton sketch with Gaussian embeddings [LP19]. We also observe similar convergence for the SRHT, which is not explained by existing worst-case theory [Tro11], but it matches the predictions based on high-dimensional asymptotic analysis of the SRHT [LLDP20]. Except for CIFAR-10, RSS and RSS-lev-scores have weaker convergence rates. This suggests that the CIFAR-10 data matrix has low coherence. Except for the high-coherence synthetic data matrix for which the convergence rate is slightly worse than $d/m$, using LESS with a uniformly random sparsifier does not affect the convergence rate. Here, we implement LESS-uniform with $d$ non-zero entries per row subsampled uniformly at random. Importantly, LESS-uniform offers significant speed-ups over other sketching matrices.

Note that some curves stop earlier than others (e.g., RRS) on the wall-clock time versus error plots, because we run the Newton sketch for each embedding for a fixed number of iterations.

## E.4 Regularized least squares and effective dimension

In Figure 5, we report the error versus number of iterations of the Newton Sketch for regularized least squares regression. These results illustrate in particular our theoretical predictions: the convergence rate of Newton-LESS is upper bounded by $d_{\text{eff}}/m$. In fact, Newton-LESS has the same convergence rate as the Newton Sketch with dense Gaussian embeddings.

## E.5 Comparison with CountSketch

In Figure 6, we perform an empirical comparison of LESS-uniform with the Sparse Johnson Lindenstrauss Transform (SJLT) with one non-zero entry per column, also known as the CountSketch [CW17]. Here, we consider the wall clock time convergence for the Newton Sketch solving a least squares task on the high-coherence synthetic matrix. We present the results alongside Gaussian and Random Row Sampling (RRS) sketches.

---

[2]We refer to the public repository for implementation details about subsampling the signal, https://github.com/WJMatthew/WESAD/blob/master/data_wrangling.py.

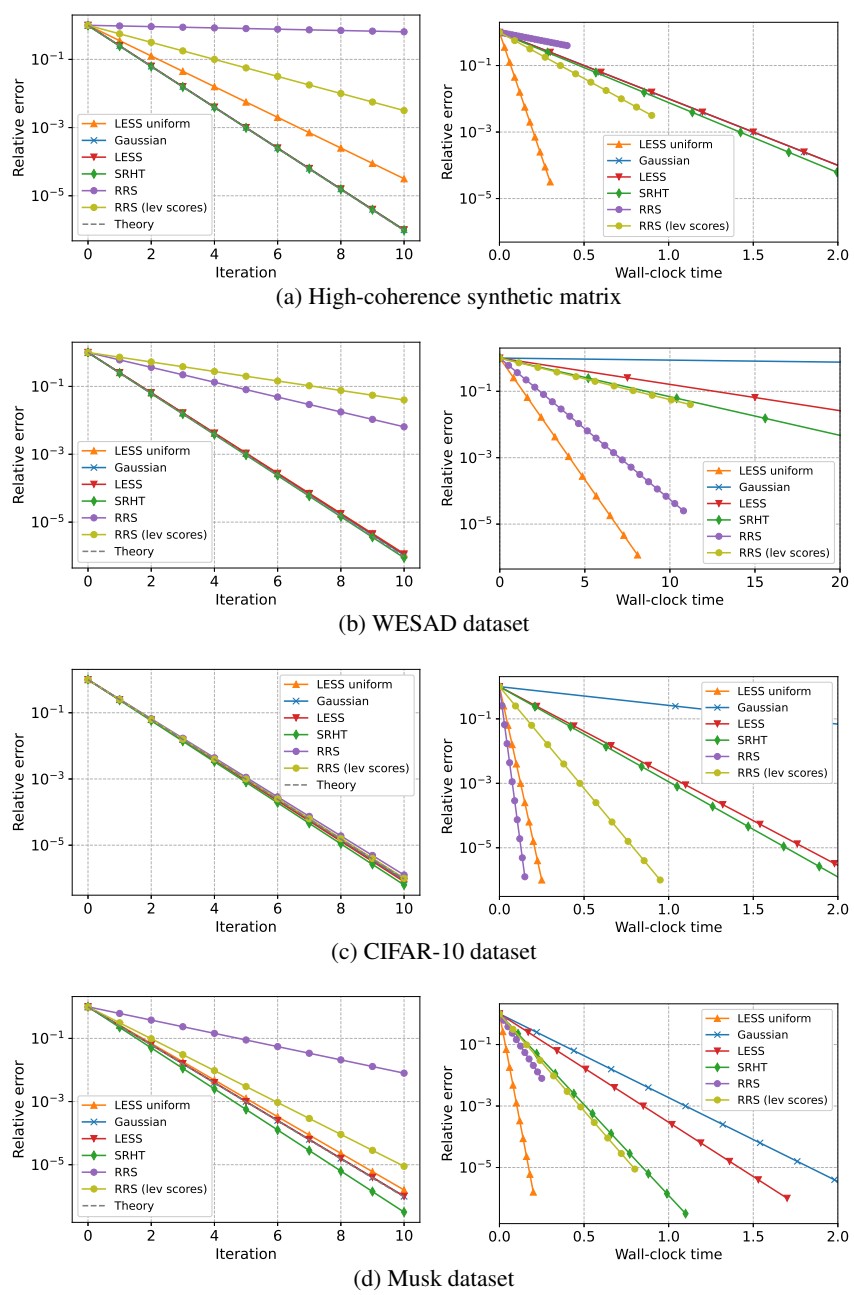

Figure 4: Newton sketch for least squares regression. We use the sketch size $m = 4d$ for all experiments. Results are averaged over 10 trials.

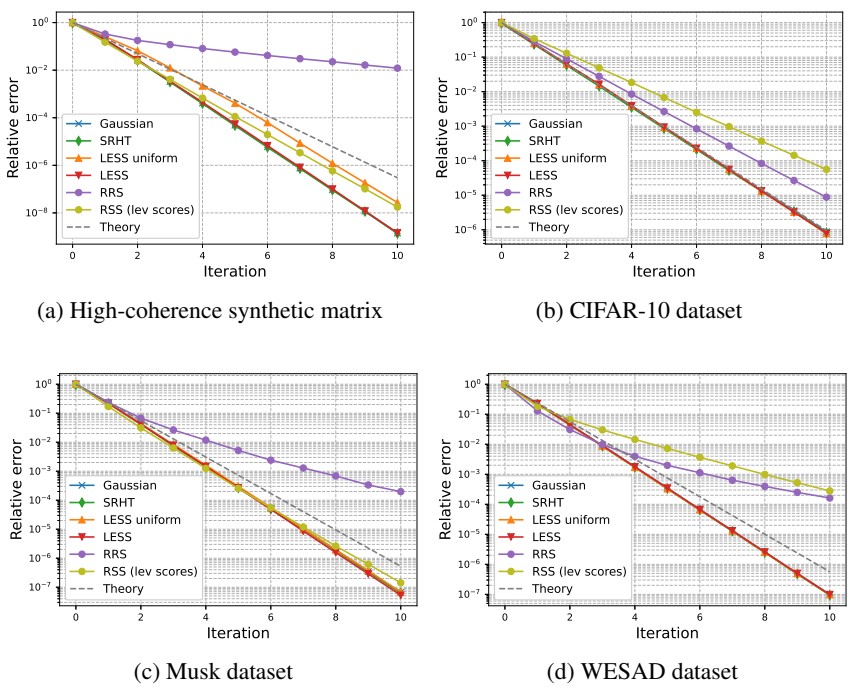

(a) High-coherence synthetic matrix

(b) CIFAR-10 dataset

(c) Musk dataset

(d) WESAD dataset

Figure 5: Newton Sketch for regularized least squares regression. We use the sketch size $m = 4d_{\text{eff}}$ for all experiments. Results are averaged over 10 trials.

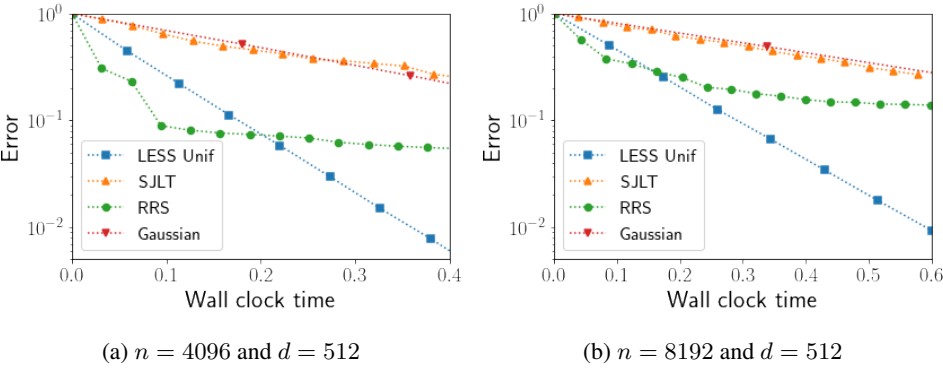

(a) $n = 4096$ and $d = 512$

(b) $n = 8192$ and $d = 512$

Figure 6: Relative error versus wall clock time: Newton Sketch for least squares regression on the high-coherence synthetic data matrix. We use sketch size $m = 1024$. For LESS-uniform, we use a number of non-zero entries $s = 0.3\,d$ per row. For SJLT, we use the standard CountSketch implementation [CW17]. Results are averaged over 10 trials.