# OpenReview forum: "Newton-LESS: Sparsification without Trade-offs for the Sketched Newton Update"
_NeurIPS.cc/2021/Conference — NeurIPS 2021 Spotlight_

### Official Review · Reviewer_WJXY · 2021-07-15

**Rating:** 7
**Confidence:** 3

**Summary:**

This work proposes a Sketched Newton method based on sparsified random Gaussian matrix. The authors provide a precious convergence analysis by a careful characterization of the first and second moment of the inverse sketched Hessian. The authors also provide empirical studies to validate the efficiency of the proposed method.

**Limitations And Societal Impact:**

The authors addressed them.

**Main Review:**

The theory is well presented and the intuition behind is well explained.

Major questions/concerns:
1. Is the work assumes the matrix $A_f(x)$ that encodes the second-order information of f given? How to obtain such encoding for a general function f efficiently?
2. As the authors argued in Section 1.2, compared to existing approaches using SRHT or Count-Sketch, the proposed Newton-LESS wins by a $Clogd$ factor, where $C$ is a non-negligible constant. Authors should compare the proposed approach with these methods as baselines in empirical studies in addition to random Gaussian and vanilla Newton.
3. Can the authors provide more insights about the matrix Q, which plays a crucial role in the analysis?

**Time Spent Reviewing:**

4

---

> ### Author Response · Authors · 2021-08-10
> **Thanks for the positive feedback! We clarify a few minor points**
>
> Thanks for the feedback! We respond to the comments below.
>
> 1. The encoding $A_f(x)$ can be obtained for functions that can be decomposed into a sum over $n$ components, which is standard in machine learning (the components are data points). For example, given any Generalized Linear Model  where $f(x) = \sum_{i=1}^n l_i(\phi_i^\top x)$, which includes logistic regression, we can let the $i$th row of $A_f(x)$ be $\sqrt{l_i''(\phi_i^\top x)}\cdot \phi_i^\top$, where $\phi_i$ corresponds to the $i$th data point.
>
> 2. In our experiments, we compared LESS to SRHT in Figure 3d  (Section 5), and to both SRHT and Leverage Score Sampling (RRS lev scores) in Figures 4, 5, 6, 7, and 11 (Appendix E). In the camera ready we will include additional experiments comparing LESS to the CountSketch. Thanks for the excellent suggestion!
>
> 3. The matrix $Q$ is essentially a sketched estimate of the inverse Hessian, but it is normalized by the Hessian so that $E[Q]\approx I$. Then, the quality of the estimate is effectively determined by its matrix variance, i.e., $E[Q^2] - E[Q]^2$ (see lines 236-253 in Section 3). The reason for the normalization is to ensure that the convergence guarantees are independent of the condition number of the Hessian.

---

### Official Review · Reviewer_1HfP · 2021-07-15

**Rating:** 6
**Confidence:** 3

**Summary:**

The paper considers using sketching to accelerate Newton's optimization method. It is argued in the paper that there exist some cases where the Hessian of the function to be optimized can be represented as A^\top A for a tall and skinny matrix A of size N \times d. The authors argue that if the cost of computing the matrix A is small and the complexity is dominated by the matrix multiplication time for computing A^\top A then approximating the Hessian with A^\top S^\top S A, for some sparse sketch matrix S, can accelerate Newton's method. The paper analyzes the convergence of the sketched Newton's method for a recently proposed sketching method known as LESS and proves a linear convergence rate (in expectation).

**Limitations And Societal Impact:**

My main concern is that the paper does not discuss at all why they had to choose LESS for sketching the Hessian and not other sketches such as subsampled randomized Hadamard transform, OSNAP, or Countsketch. The only advantage of LESS compared to other sketches is having a small inversion bias. More specifically, sketching the Hessian using LESS and then inverting the sketched matrix can achieve an error (or bias) of \eps = 1/\sqrt{d} as opposed to a constant error. It is not clear at all why a constant error like \eps = 1/100 would not suffice for the convergence proof.
Any sketch matrix that satisfies the subspace embedding property with some constant error parameter like \eps = 1/100 would spectrally approximate the Hessian and thus can be used to spectrally approximate the inverse Hessian up to a multiplicative (1 \pm \eps) factor. It seems to me that this is enough for the convergence proof and there is no need for having \eps = 1/\sqrt{d}.
Note that the convergence rate in Theorem 1 and 4 are (d/m * (1+\eps))^T. The 1+\eps factor does not matter here and as long as m is a constant factor larger than d, a linear convergence rate can be established. Can the authors please explain why it is crucial to have  \eps = 1/\sqrt{d}?

**Main Review:**

The paper considers using sketching to accelerate Newton's optimization method. It is argued in the paper that there exist some cases where the Hessian of the function to be optimized can be represented as A^\top A for a tall and skinny matrix A of size N \times d. The authors argue that if the cost of computing the matrix A is small and the complexity is dominated by the matrix multiplication time for computing A^\top A then approximating the Hessian with A^\top S^\top S A, for some sparse sketch matrix S, can accelerate Newton's method. The authors mainly focus on a sketch matrix S which is recently proposed and is known as LESS. The LESS is a data-dependent sketch that (roughly speaking) is an average of d independent sampling matrices, where each sampling matrix samples the rows of A according to their leverage scores. The paper analyzes the convergence of the sketched Newton's method and proves a linear convergence rate (in expectation).

I was wondering if the authors could demonstrate that the assumption that the Hessian is in the form of A^\top A holds in any popular ML problem besides linear regression. For instance, is the Hessian for Logistics regression in this form?

My main concern is that the paper does not discuss at all why they had to choose LESS for sketching the Hessian and not other sketches such as subsampled randomized Hadamard transform, OSNAP, or Countsketch. The only advantage of LESS compared to other sketches is having a small inversion bias. More specifically, sketching the Hessian using LESS and then inverting the sketched matrix can achieve an error (or bias) of \eps = 1/\sqrt{d} as opposed to a constant error. It is not clear at all why a constant error like \eps = 1/100 would not suffice for the convergence proof.
Any sketch matrix that satisfies the subspace embedding property with some constant error parameter like \eps = 1/100 would spectrally approximate the Hessian and thus can be used to spectrally approximate the inverse Hessian up to a multiplicative (1 \pm \eps) factor. It seems to me that this is enough for the convergence proof and there is no need for having \eps = 1/\sqrt{d}.
Note that the convergence rate in Theorem 1 and 4 are (d/m * (1+\eps))^T. The 1+\eps factor does not matter here and as long as m is a constant factor larger than d, a linear convergence rate can be established. Can the authors please explain why it is crucial to have  \eps = 1/\sqrt{d}?

**Time Spent Reviewing:**

4

---

> ### Author Response · Authors · 2021-08-10
> **Reviewer seems to have several misunderstandings about our technical contributions, which we aim to clarify**
>
> Thanks for the feedback! It seems that much of the concern in the review is based on the claim that, other than small inversion bias, LESS embeddings do not provide any advantages over existing sketches. On the contrary, our main convergence results for LESS embeddings are fundamentally stronger and more precise than those achieved by prior work for other sketching methods (such as SRHT, OSNAP, CountSketch). We establish not only linear convergence but also the precise rate of this convergence, and this rate is strictly better than is known for other sketches. These improvements are not based on the usual subspace embedding arguments (the usual way to analyze SRHT, OSNAP, CountSketch, etc.), or on small inversion bias of LESS, but rather on new properties of LESS that we develop in this paper. To our knowledge, no other sketching method, including SRHT, OSNAP, CountSketch, satisfies these stronger results. While we address the comparison to other sketching techniques in several sections of the paper (e.g., Sections 1.2, 2 and 5), we will substantially clarify this in the camera ready. Below, we respond to the detailed comments from the review. Hopefully, after considering our responses, the reviewer is comfortable increasing the score to be more in line with the other reviewers.
>
> 1. *"is the Hessian for Logistics regression in this form"*: The Hessian can be easily decomposed into $A^\top A$ for any Generalized Linear Model, i.e., where $f(x) = \sum_{i=1}^n l_i(\phi_i^\top x)$, which includes logistic regression (this is well known in the literature, see [PW17,WRKXM18]). In fact, we use logistic regression for our experiments in the paper (see Figure 3 in Section 5, and Figures 8, 9, 10 in Appendix E).
>
> 2. *"the paper does not discuss at all why they had to choose LESS"*: We disagree. Our new results for LESS are compared to other state-of-the-art sketching methods (SRHT, OSNAP, CountSketch, Leverage Score Sampling) in several sections of the paper: (a) in terms of convergence guarantees in Section 1.2; (b) in terms of computational cost in Section 2; and (c) in terms of empirical performance in Section 5 and Appendix E (for SRHT and Leverage Score Sampling). Our key motivation for using LESS is so that we can establish a better convergence rate than any of the aforementioned methods, namely, $(\frac dm)^T$ versus $(C\log d\cdot \frac dm)^T$ where $C>1$ is a non-negligible constant (see lines 175-179 in Section 1.2).
>
> 3. *"The only advantage of LESS compared to other sketches is having a small inversion bias"*: We strongly disagree, since small inversion bias is not sufficient to obtain our new convergence guarantees. As discussed in lines 160-165 (Section 1.2) and Appendix D, small inversion bias of LESS (i.e., near-unbiasedness of the Newton estimates) is only useful in the distributed setting, whereas our main results focus on showing improved convergence rates for the standard (non-distributed) Newton Sketch using techniques that go beyond small inversion bias, such as characterizing the second moment of the inverse Hessian estimate (see Section 3).
>
> 4. *"Any sketch matrix that satisfies the subspace embedding property... is enough for the convergence proof"*: That is true, if one simply wants to establish linear convergence. However, to obtain optimal performance, it is not enough to establish linear convergence, but rather it is essential to characterize the rate of this convergence. For example, in our experiments (Figures 8, 9, 10 in Appendix E), simple gradient descent (GD) exhibits linear convergence, and yet it is several orders of magnitude slower than the Newton Sketch based methods. Also, different choices of sketching matrices in the Newton Sketch lead to substantially different linear convergence rates (Figures 4, 5, 6, 7, and 11 in Appendix E), which affects the optimization time. Crucially, our results are some of the first to obtain convergence rates that are strictly better than what can be shown just by relying on the subspace embedding property (see lines 175-179 in Section 1.2).
>
> 5. *"Can the authors please explain why it is crucial to have $\epsilon = 1/\sqrt{d}?$"*: We are happy to, since this is an important technical point that we don't want to be lost on any reader. The main message of our paper is that the Gaussian sketch can be sparsified *without trade-offs* by using LESS embeddings. To that end, we show (see Theorem 1 and Remark 2) that the convergence rate of Newton-LESS is the same as that of Gaussian Newton Sketch, namely $(\frac dm)^T$, up to lower order terms. The "lower order terms" claim is justified by the fact that $\epsilon$ vanishes when the dimension $d$ is large (specifically, $\epsilon=O(1/\sqrt d)$), which means that for high-dimensional problems the difference in convergence rates is negligible. In contrast, for other methods the convergence rate is only upper bounded by $(C\log d \cdot \frac dm)^T$ (i.e., $\epsilon=O(\log d)$), so there is a non-negligible convergence trade-off compared to using the Gaussian sketch.

---

> > ### Comment · Reviewer_1HfP · 2021-08-14
> > **Respond to Rebuttal**
> >
> > I appreciate the authors' feedback.
> > They have addressed my confusion.
> > The paper improves the convergence rate of sketches with subspace embedding guarantees such as SRHT and OSNAP by a factor of $\log d$ which is quite interesting. Therefore, I raise my score to 6.

---

### Official Review · Reviewer_Stjd · 2021-07-16

**Rating:** 8
**Confidence:** 3

**Summary:**

This paper analyzes a recently proposed sketching transform (LESS) that has been suggested for Newton Sketch. LESS (originally proposed in [DLDM20]) is aimed to be between full Gaussian sketch and row sampling. It achieves that by sampling the Gaussian sketch, with a sampling scheme based on the leverage scores. In the context of Newton Sketch, Gaussian sketch has been shown to work well, but using is mostly useless since the sketch operation is as expensive as computing the Hessian. Other sketches (e.g. SRHT) reduce per iteration cost, but increase the number of iterations. With LESS, the authors show we can avoid this tradeoff for certain types of functions.

The convergence rate the authors prove is linear (and not superlinear as one can hope from a second order method). But crucially, the rate is independent of the condition number.

The authors also consider the special case of ridge regression.

**Limitations And Societal Impact:**

No limitations and societal impact

**Main Review:**

Overall, I think this is a very good paper. It analyzes an important method (Newton Sketch), and shows how to use a specific sketch to make it more attractive. The paper is very nicely written.

I have several (mostly minor) comments:

1) Line 20: contrary to what this sentence and the next says, the paper does not handle any twice diff function, but only one that can be written as a finite sum of many simple functions.

2) Theorem 1/4/10: I am confused about the role of epsilon. Does not seem that epsilon come into play in the various bounds in the theorem statement. Is it a parameter that is chosen, or is it fixed? If it is chosen: what does epsilon O(1/\sqrt{d}) mean?  If it is fixed: on what does the constant in O(1/sqrt{d}) depend?

3) Doesn't line 124-125 contradict remark 2? Remark 2 says the bound was known for Gaussian sketch, but line 125 says it was known only for the case of least squares.

4) Definition 2: can LESS requirements be stated with overestimates to leverage scores, with sampling size increased by how much severe the overestimate is? Such framing is common in literature that uses leverage score sampling, and will enabled defining LESS and LESS-uniform in a single definition.

5) Condition 2: What does alpha=O(1) mean here?

6) What is the even Epsilon that the various results (Theorem 1/4/10) depend on? Is it possible to had a safeguard in the algorithm to check if the even holds, and if not, generate a new sketch?

**Time Spent Reviewing:**

6

---

> ### Author Response · Authors · 2021-08-10
> **Thanks for the positive feedback! We clarify a few minor points**
>
> Thanks for the feedback! We respond to the comments below.
>
> 1. Regarding line 20: That's right, we consider specifically functions that can be written as a sum of simple functions.
>
> 2. Regarding epsilon: This is not a parameter that is chosen, but rather, it is an approximation error in our estimate of the convergence rate, and the constant in $O(1/\sqrt d)$ is absolute. For instance, Theorem 1 states that the convergence rate (left-hand-side) is between $(1-c/\sqrt d)\frac dm$ and $(1+c/\sqrt d)\frac dm$ for an absolute constant $c$.
>
> 3. Regarding line 124-125 and Remark 2: To our knowledge, the precise convergence rate (i.e., down to lower order terms) for the Gaussian Sketch was previously derived only for least squares (and it does not immediately extend to more general losses). As discussed in Section 4, our convergence analysis applies not just to Newton-LESS, but to a broad class of sketching methods which includes Gaussian Sketches (see Theorem 10), so this is what we point out in Remark 2.
>
> 4. Regarding Definition 2: Yes, in fact the sufficient condition is that $p_i = \Omega(l_i(A,C)/s)$, where $s$ is the number of non-zeros per row (see Appendix C), which unifies LESS and LESS-uniform. We make the distinction in Definition 2 simply to highlight the two most relevant implementations.
>
> 5. Regarding Condition 2: The expression $\alpha=O(1)$ means that $\alpha$ is bounded by some absolute constant (e.g., by 2). The specific value of the constant will only affect the other absolute constants used in the main results, for example in Theorem 6. We use this notation to highlight that no particular choice of these constants is significant for the analysis.
>
> 6. Regarding event $\mathcal E$: This event ensures that the Hessian estimate is sufficiently well-conditioned (relative to the true Hessian). In all our results, this event holds with very high probability. In principle, it would be possible to check it, but that would require computing the exact Hessian, so it would be impractical. Also, in our empirical results we show that our theoretical convergence estimates are extremely accurate, so the effect of conditioning on the event is negligible in practice.

---

> > ### Comment · Reviewer_Stjd · 2021-08-22
> > **Thank you for the response!**
> >
> > Thank you for the response! I have read it, and my accept recommendation remains.

---

### Official Review · Reviewer_nNxo · 2021-07-18

**Rating:** 7
**Confidence:** 3

**Summary:**

The paper studies a method to speed up Newton-type optimization methods using randomized sketching techniques (specifically, leverage score sparsification-based sketching).

**Main Review:**

Newton-type optimization methods are hampered by their expensive computational costs (primarily arising due to the need to compute an inverse Hessian in every iteration). To alleviate this cost, a possible solution path is to randomly sketch/embed (the factors of) the Hessian into a lower number of components. Several schemes following this path have been proposed, including subsampling-based sketches and Gaussian sketches. The former (subsampling) reduces computation but affects convergence, while the latter enjoy strong convergence rates but does not benefit costs.

This paper shows that sketching based on leverage-score sparsification ("LESS embedding") achieves nearly optimal computation, but also retains certain key moment properties of Gaussian sketches and therefore achieve matching problem-independent convergence rates.

The paper is very well written and the contributions are clearly justified with theory and experiments.

Some comments:
- "convergence rate and computation costs are both nearly at their optimal values." Unclear what this means exactly. Are there known lower bounds, and if so, can the authors provide a citation? If not, this claim should be justified.  In any case, what does "nearly" mean in this claim? The mixed rate in Eq 4 may suggest the existence of better techniques.
- I'm somewhat puzzled by the "T" parameter in Theorems 1,4 and Corollary 3.  Is T an arbitrary positive integer? Specifically, the sketch size depends on T, which is counter-intuitive (one would imagine a convergence result to be stated in terms of "for all t > T, XYZ holds").
- Consider increasing legend sizes in Fig 2 and 3, they are very difficult to parse.

---
Thanks for the responses, and good luck.

**Time Spent Reviewing:**

6

---

> ### Author Response · Authors · 2021-08-10
> **Thanks for the positive feedback! We clarify a few minor points**
>
> Thanks for the feedback! We respond to the comments below.
>
> 1. Regarding the convergence rate and computational cost being "nearly at their optimal values": We agree that this statement is somewhat imprecise in its current form, but we were trying to convey the intuition before we got more precise. Here, by "optimal" we mean "when optimized over the sparsity of the sketch". In this context, for the convergence rate, the sparsity is optimized by the dense sketch ($n$ non-zeros per row), whereas the computational cost is optimized by the row sampling sketch ($1$ non-zero per row). As explained in Section 1.1, we show that Newton-LESS achieves the same convergence rate as a dense sketch, up to lower order terms (i.e., nearly the same). Also, at the end of Section 2, we explain that the computational cost of Newton-LESS is up to logarithmic factors the same as the cost of Newton Sketch with row sampling, which is $O(nnz(A) + md^2)$.
>
> 2. Regarding the parameter $T$: This parameter represents the number of iterations performed by the Newton Sketch algorithm. It is true that the required sketch size m grows with $T$ (albeit only logarithmically), and this is in fact unavoidable in general. At each iteration, we perform a sketch of the Hessian, and so there is a very small but non-zero chance that we obtain an extremely ill-conditioned estimate which would arbitrarily interfere with the convergence (this is why we use conditional expectation in our statements). As $T$ grows large, the probability that one of the sketches is extremely ill-conditioned gradually increases. However, we show that this probability is negligibly small, and this issue is generally not observed in practice.
>
> 3. Thanks for pointing out the small legend sizes, we will increase them for the camera ready.

---

### Author Response · Authors · 2021-08-10
**Thanks to all reviewers for their comments**

Three reviewers rated the paper highly, either accept or strong accept.  One rated the paper much lower, as a reject.  The reasons for this are based on misunderstandings which we attempt to clarify.  We reply to each reviewer in turn.

---

### Decision · Program_Chairs · 2021-09-27

**Decision:**

Accept (Spotlight)

**Comment:**

The reviewers all appreciated the improved benefits of the new dimensionality reduction scheme over existing methods for achieving subspace embeddings. There was initially some confusion about why existing methods could not achieve similar results, as well as to which second order optimization methods the method could be applied to, but these were clarified well in the rebuttal, and all authors were in agreement and enjoyed reading this paper.